# An improved global pressure and ZWD model with optimized vertical correction considering the spatial-temporal variability of multiple height scale factors

Chunhua Jiang[1,2,3], Xiang Gao[1], Huizhong zhu[1], Shuaimin Wang[4], Sixuan Liu[1], Shaoni Chen[1], Guangsheng Liu[1]

[1]School of Geomatics, Liaoning Technical University, Fuxin, 123000, China
[2]State Key Laboratory of Geo-Information Engineering, Xi'an, 710054, China
[3]State Key Laboratory of Geodesy and Earths' Dynamics, Innovation Academy for Precision Measurement Science and Technology, CAS, Wuhan, 430077, China
[4]College of Mining and Geomatics, Hebei University of Engineering, Handan, 056038, China

*Correspondence to*: Chunhua Jiang (jiangchunhua@sdu.edu.cn), Xiang Gao (472120799@stu.lntu.edu.cn), Huizhong zhu (zhuhuizhong@lntu.edu.cn)

**Abstract.** Atmospheric pressure and Zenith wet delay (ZWD) are essential for GNSS tropospheric correction and precipitable water vapor (PWV) retrieval. As the development progresses of real-time GNSS kinematic technology, moving platforms such as airborne and shipborne require high-quality tropospheric delay information to pre-correct errors. Most existing tropospheric models are only applicable to the Earth surface, while exhibiting poor accuracies in high-altitude areas due to simple vertical fitting functions and limited temporal resolution of the underlying parameters. Hence, an improved global empirical pressure and ZWD model is developed using 5-years ERA5 hourly reanalysis data, called IGPZWD, which takes seasonal and intraday variations into consideration. The vertical accuracy and applicability of IGPZWD model are further optimized by introducing the annual and semi-annual harmonics for pressure and ZWD height scale factors of exponential function with three orders. Taking the ERA5 and radiosonde profiles data in 2020 as reference, the pressure and ZWD of IGPZWD model show superior performance than those of three state-of-the-art models, i.e., GPT3, IGPT and GTrop. Furthermore, IGPZWD-predicted ZTD yields improvements of up to 65.7%, 2.4% and 7.8% over that of GPT3, RGPT3 and GTrop models on a global scale respectively. The proposed vertical correction algorithm effectively weakens the impact of accumulation error caused by excessive height difference, achieving optimal accuracy and feasibility in the high-altitude area. The IGPZWD model can be extensively applied in GNSS kinematic precision positioning as well as atmospheric water vapor sounding.

## 1 Introduction

Tropospheric delay is a typical error in the application of microwave-based space-geodetic techniques. (Hofmeister and Böhm, 2017; Xu et al., 2023; Lu et al., 2023). In the field of global navigation satellite systems (GNSS), zenith tropospheric

delay (ZTD) is correlated to station coordinates and receiver clock error (Li et al., 2023). Accurate external prior ZTD can effectively improve positioning precision and enhance convergence speed (Tregoning and Herring, 2006; Sun et al., 2019; Zhang et al., 2022). Besides, the troposphere contains diverse atmospheric information. Accurate precipitable water vapor (PWV) can be derived by the combination of ZTD, atmospheric pressure and weighted mean temperature, and applied as an important indicator for regional and global numerical weather forecasting and meteorological monitoring (Wang et al., 2016; Li et al, 2022). In general, the slant path delay (SPD) of the GNSS signal is divided into hydrostatic delay and non-hydrostatic (wet) delay components, each of which can be expressed as the multiplication of the zenith delay and mapping function (Landskron and Böhm, 2017). The zenith hydrostatic delay (ZHD) can be accurately determined according to the Saastamoinen model with measured instantaneous pressure as the input, while the zenith wet delay (ZWD) is generally estimated as an unknown parameter (Saastamoinen, 1972., Hadas et al., 2017., Zhang et al., 2021., Yang et al., 2023). Hence, accurate pressure and ZWD are crucial prerequisites for obtaining reliable tropospheric delay prior information.

Generally, accurate pressure, temperature and humidity observations can be obtained from meteorological instrument. But most GNSS stations are not equipped with meteorological sensors, and the spatial distribution of automatic weather stations can't meet the growing demands of high-precision positioning. Numerical weather models (NWM) provide high-quality reanalysis products, but these atmospheric data come with release latency and heavy storage burden (Zhang et al., 2019; Su et al., 2021). As a trade-off, multiple empirical models have been constructed using historical reanalysis data, which can predict tropospheric parameters for real-time GNSS applications (Schueler et al., 2001; Leandro et al., 2006, 2008; Boehm et al., 2007; Lagler et al., 2013, Böhm et al., 2015; Landskron and Böhm, 2017). Unfortunately, such models mainly incorporate annual and semi-annual harmonics to reflect the long-term pattern in parameters, which makes it difficult to capture short-term fluctuations. Therefore, some scholars have developed the models which introduced the diurnal and semi-diurnal terms using hourly reanalysis data, such as the TropGrid2 model (Schüler, 2014), the ITG model (Yao et al., 2014), the WHU_CPT (Zhang et al., 2018) and IGPT models (Li et al., 2021), etc.

To reduce the accuracy loss caused by the height difference between model grid and target position, multiple fitting functions are used to simulate the vertical nonlinear variations of pressure and ZWD. Regarding the pressure, typical vertical correction methods include the original and modified standard extrapolation model (Berg, 1948; Su et al., 2021), the hydrostatics and ideal gas equation (Wang et al., 2007), the exponential model related to the virtual temperature (Yao et al., 2014; Böhm et al., 2015) and the adiabatic model based on the temperature lapse rate (Benjamin and Miller, 1990; Mao et al., 2021; Sun et al., 2023). ZWD exhibits complex vertical variation due to the dynamic nature of water vapor. Exponential fitting function with a single decay coefficient is typically applied for the vertical correction of ZWD, but some studies reveal that the ZWD height scale factor shows obvious regional differences and periodic characteristics (Kouba, 2008; Sun et al., 2017; Yao et al., 2018; Huang et al., 2021a; Huang et al., 2023). Accordingly, substantial efforts have been made to construct optimized regional and global ZWD vertical correction models which take seasonal variation or long-term linear trend into consideration, for instance, the GTrop model (Sun et al., 2019), the GZWD-H model (Huang et al., 2021b), the TZ (Xu et al., 2023) and HPZI models (Zhao et al., 2024), etc. Furthermore, the piecewise function and stratification methods

have been verified to be applicable and feasible for the vertical correction of ZWD (Li et al., 2015; Yao et al., 2018; Hu and Yao, 2019; Zhu et al., 2022).

With the development of GNSS infrastructure, moving platforms such as unmanned aerial vehicle (UAV), shipborne and moving vehicles provide massive spatial data for navigation and positioning (Wang et al., 2022; Zhang et al., 2023).
Nonetheless, the complex weather and geographical condition, insufficient meteorological data and limited satellite observation geometry pose great challenges to kinematic GNSS solutions (Rocken et al., 2005; Webb et al., 2016; Penna et al., 2018). Generally, accurate and reliable tropospheric delay constraints can effectively enhance the performance of positioning. Xia et al. (2023) comprehensively considered the seasonal and intraday variations of the elevation normalization factor and developed a real-time ZTD model, and the vertical convergence speed was improved by 37.4% after the ZTD
constraints are utilized to the float precise point positioning (PPP). Besides, FL-ZTD and SL-ZTD models are established using the piecewise exponential function as the key vertical adjustment scheme for ZTD, which reduced the convergence time by 60.0% and 33.3% compared to the standard PPP, respectively (Zhang et al., 2020). An optimized GPT3 model (RGPT3) is constructed using random forest (RF), achieving 12.3% and 7.9% improvement in vertical convergence speed and accuracy (Li et al., 2023). However, most of the current tropospheric models are only applicable to the Earth surface.
Although some tropospheric vertical profile models perform well in the high-altitude areas, there are still specific shortcomings such as insufficient periodic terms, fixed application scenarios and limited vertical accuracy.

To overcome above drawbacks, an empirical global pressure and ZWD grid model with broader operating space named IGPZWD is constructed using ERA5 hourly data from 2015 to 2019 in this contribution. Initially, the annual, semi-annual, diurnal and semi-diurnal periods of atmospheric pressure and ZWD are taken into consideration. Thereafter, Optimal
exponential fitting function with three orders is introduced as core vertical correction scheme. Finally, the height scale factors are estimated by least-squares algorithm refined up to semi-annual harmonics. Furthermore, the accuracy and reliability of IGPZWD are comprehensively evaluated and validated against ERA5 and radiosonde data in 2020.

## 2 Data and methodologies

### 2.1 Data sources

The fifth generation European Centre for Medium-Range Weather Forecasts (ECMWF) atmospheric reanalysis (ERA5) benefits from four-dimensional variational (4D-Var) assimilation solution and integrated forecasting system (IFS) forecast systems, which provides high spatial-temporal resolution and high-accuracy atmospheric state variables over globe (Hersbach et al., 2020). ERA5 provides 3D pressure-level products with a vertical resolution of 37 levels and 2D single-level data. The atmospheric parameters are provided with a horizontal resolution of 0.25°×0.25°, and the hourly data can more
accurately reflect the short-term variation of meteorological parameters (Jiang et al., 2023). In this contribution, ERA5 hourly temperature, pressure, specific humidity and geopotential data on pressure-level, surface pressure, 2m-dewpoint

temperature and 2m-temperature on single-level from 2015 to 2019 are utilized to construct the IGPZWD model, and the accuracy of the new model is verified using data in 2020.

The Integrated Global Radiosonde Archive (IGRA) consists of radiosonde and pilot balloon observations from more than 2800 globally distributed stations, and surface and upper-air meteorological data become available in near real-time from about 800 stations worldwide (Ingleby et al., 2016). Atmospheric temperature, pressure and water vapor pressure data profiles at 0:00 coordinated universal time (UTC) and 12:00 UTC in 2020 are obtained from the IGRA. Generally, sensor quality and weather events have a serious impact on raw measurements, which result in missing data and outliers. Hence, the low-quality radiosonde data profiles which meet the following quality control standards are eliminated. (1) The height difference between two successive levels is greater than 2 km. (2) The pressure difference between two successive levels is greater than 200 hPa. (3) The height of the top-level data is less than 10 km. (4) The effective observation records of the profile are less than 20. Finally, the geographical distribution of selected 565 radiosonde stations is presented in Figure 1.

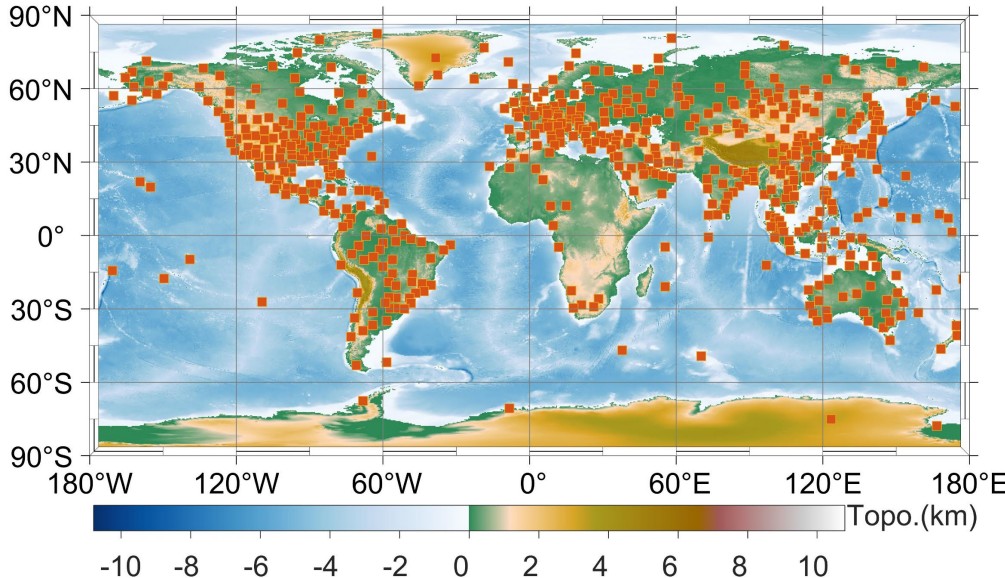

Figure 1: The geographical distribution of selected 565 radiosonde stations.

## 2.2 Inversion Strategies for ZWD and ZTD

The ERA5 and radiosonde ZWD profiles are calculated according to the numerical integration method as follows (Thayer, 1974; Askne and Nordius, 1987; Jiang et al., 2023):

$$e = \begin{cases} PQ/(0.378 \times Q + 0.622) & Pressure-level \\ 6.112.\exp(17.62 \times Dew/(243.12 + Dew)) & Single-level \end{cases} \tag{1}$$

$$N_W = K_2' \times e/T + K_3 \times e/T^2 \tag{2}$$

$$115 \quad ZWD = \int_{H_0}^{Htop} N_W \, dH \tag{3}$$

Where $K_2' = 22.97$ K/hPa, $K_3 = 375463$ K²/hPa. $H^{top}$ and $H_0$ are the heights of the top and bottom levels of the parameter profile. $Q$, $P$, $T$, $Dew$ and $e$ are the specific humidity, pressure, temperature, 2m-dewpoint temperature and water vapor pressure for each level, respectively. The radiosonde ZTD profile is derived by a combination of Saastamoinen model and integration method (Fernandes et al., 2021). The specific process is as follows:

$$120 \quad N_T = K_1 \times (P - e) + K_2 \times e / T + K_3 \times e / T^2 \tag{4}$$

$$ZHD^{top} = \frac{0.0022768 \cdot P^{top}}{1 - 0.00266 \cos 2\varphi - 0.00028 H^{top}} \tag{5}$$

$$ZTD = ZHD^{top} + \int_{H_0}^{H^{top}} N_T \, dH \tag{6}$$

Where $K_1 = 77.604$ K/hPa, $K_2 = 64.79$ K/hPa. $P^{top}$ is the pressure of top level, and $\varphi$ is the grid latitude in rad. $ZHD^{top}$ denote the zenith hydrostatic delay above the top level, which is added to the integral ZTD, ensuring the accuracy of
125 radiosonde ZTD as a reference value (Huang et al., 2023; Fan et al., 2020).

## 3 Development of IGPZWD model

### 3.1 The spatial-temporal variation characteristics of surface pressure and ZWD

To reasonably account for the spatial-temporal dependency of the pressure and ZWD, the annual mean values, annual, semi-annual, diurnal and semi-diurnal amplitudes of global ERA5 surface pressure and ZWD from 2015 to 2019 are determined
by least-squares algorithm, which are surfaced as presented in figure 2. The annual mean pressure in high-altitude areas such as Greenland, Tibet Plateau and Antarctica are generally small due to low atmosphere density. The annual and semi-annual amplitudes in the middle and high latitudes are higher than those in the low latitudes. The geographical distribution of the diurnal and semi-diurnal amplitudes is opposite, indicating strong intraday variations of pressure in the low latitudes. The magnitude of ZWD is positively correlated with atmospheric water vapor content, resulting in higher annual mean values in
the tropics characterized by high temperature and abundant rainfall. Correspondingly, the ZWDs exhibit strong seasonal and intraday variations in these areas. Evident annual and semi-annual amplitudes of ZWD are observed in the southern North America, northern Africa, and southeastern Asia, where the corresponding values exceed 50 mm. Additionally, ZWDs exhibit relatively high diurnal and semi-diurnal amplitudes in the tropical coastal area and these intraday variations can't be fully absorbed by the seasonal signal residuals in the modeling. It is demonstrated that the temporal variations of pressure

and ZWD mainly depend on geolocation, and the intraday periods can't be ignored. Therefore, a regular 1° grid is chosen to simulate the spatial variations of pressure and ZWD, and the following harmonic function is employed to account for the seasonal and intraday variations of the two parameters at each grid point.

$$P_r(ZWD_r) = a_0 + \sum_{l=1}^{2}\left[a_l\sin\left(\frac{2\pi l.\text{DOY}}{365.25}\right)+b_l\cos\left(\frac{2\pi l.\text{DOY}}{365.25}\right)\right]+\sum_{l=1}^{2}\left[c_l\sin\left(\frac{2\pi l.\text{HOD}}{24}\right)+d_l\cos\left(\frac{2\pi l.\text{HOD}}{24}\right)\right] \quad (7)$$

Where $a_0$ is the annual mean value of pressure or ZWD. $a_l$, $b_l$, $c_m$ and $d_m$ are the coefficients of annual, semi-annual,

diurnal and semi-diurnal periodic terms, respectively. DOY and HOD denote the "day of the year" and the "hour of the day", respectively.

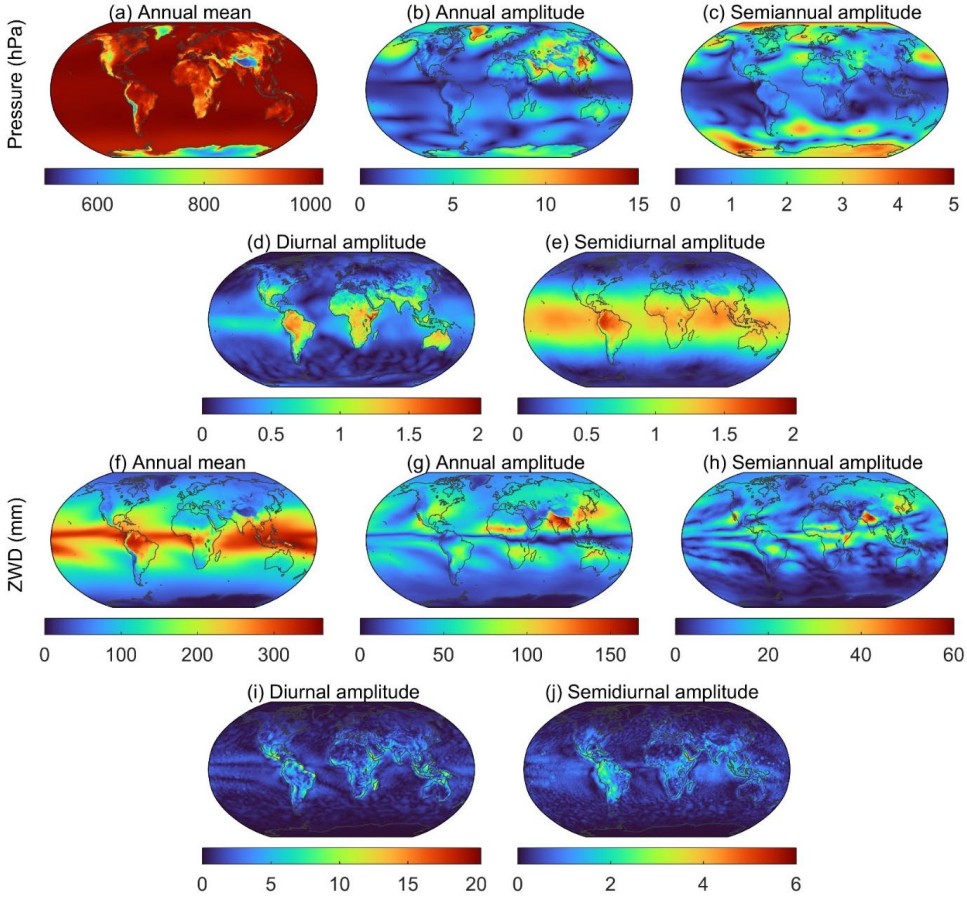

Figure 2: The annual mean values, annual amplitude, semi-annual amplitude, diurnal and semi-diurnal amplitudes of global ERA5 surface pressure (a-e) and ZWD (f-j) from 2015 to 2019. Note that the colorbar scales of each subgraph are different.

## 3.2 Vertical fitting algorithm for pressure and ZWD

With the assumption of hydrostatic equilibrium, the dry-air differential equation for the determination of pressure based on temperature is expressed as follows (Kleijer, 2004):

$$\frac{1}{P}dP = \frac{-g_m}{R_d T}dH \tag{8}$$

Where $g_m$ denotes mean gravity acceleration, $R_d = 287.06 \pm 0.01$ J$^{-1}$kg$^{-1}$. The temperature is nearly linear with height in the troposphere and stratosphere, and thus the lapse rate ($\lambda$) can be regarded as a constant value over a short vertical range. Substituting $dH = -dT/\lambda$ into equation (8), the pressure corresponding to the temperature at the sea-level height ($T_0$) and target height ($T_h$) can be expressed as follows:

$$P_h = (T_h)^{\tau+1}, P_0 = (T_0)^{\tau+1}, \tau = \frac{g_m}{R_d \lambda} - 1 \tag{9}$$

Where $T_h = (T_0 - \lambda h)$, and thus the above equation is integrated as:

$$P_h = P_0 \cdot \left(\frac{T_0 - \lambda h}{T_0}\right)^{\tau+1} \equiv P_0 \cdot \exp\left[(\tau+1)\cdot\ln\left(1 - \lambda h / T_0\right)\right] \tag{10}$$

Based on Taylor series expansion, equation (11) can be further expressed as:

$$P_h = P_0 \cdot \exp(\sum_1^n \beta_{Pn} h^n), \beta_{Pn} = -(\tau+1)\cdot(\lambda/T)^n / n \tag{11}$$

According to the study of Wang et al. (2022), the vertical ZWD profiles can also be accurately fitted using a multi-order exponential function, and the corresponding equation is as follow:

$$ZWD_h = ZWD_0 \cdot \exp(\sum_1^n \beta_{Wn} h^n) \tag{12}$$

Where $ZWD_0$ and $ZWD_h$ are the ZWD at sea-level and certain height above sea-level ($h$), respectively. $\beta_{Pn}$ and $\beta_{Wn}$ denote the $n$th order height scale factors of pressure and ZWD, respectively, which are determined by nonlinear least-squares algorithm to achieve vertical correction without temperature as input.

The accuracies of pressure and ZWD fitted by exponential functions with the orders of one to four are investigated to determine the optimal one. The fitting results and residual profiles of six grid points at different geolocations are illustrated in Figure 3. Evidently, the EFO1 struggles to simulate the nonlinear vertical variation of pressure. It generally

underestimates the pressure in the range of 3-6 km, and the surface residuals even exceed 15 hPa. The EFO2 improves the fitting effect compared to the EFO1, but exhibiting large residuals at the grid points of 20.5°N, 120.5°W and 50.5°N, 120.5°W. Notably, the EFO3 and EFO4 exhibit optimal performance and small vertical residuals which stay within ±2 hPa.

Regarding the ZWD, the fitting residuals of EFO1 are obviously large below 3 km and exceed 70 mm at the grid point of 0.5°S, 120°E. In the lower troposphere, the absolute residuals of EFO2 at the three tropical grid points are still more than 30 mm, whereas the residuals of EFO3 and EFO4 stay within ±15 mm at all six grid points. Above results demonstrate that EFO3 and EFO4 can accurately capture complicated vertical variations of pressure and ZWD.

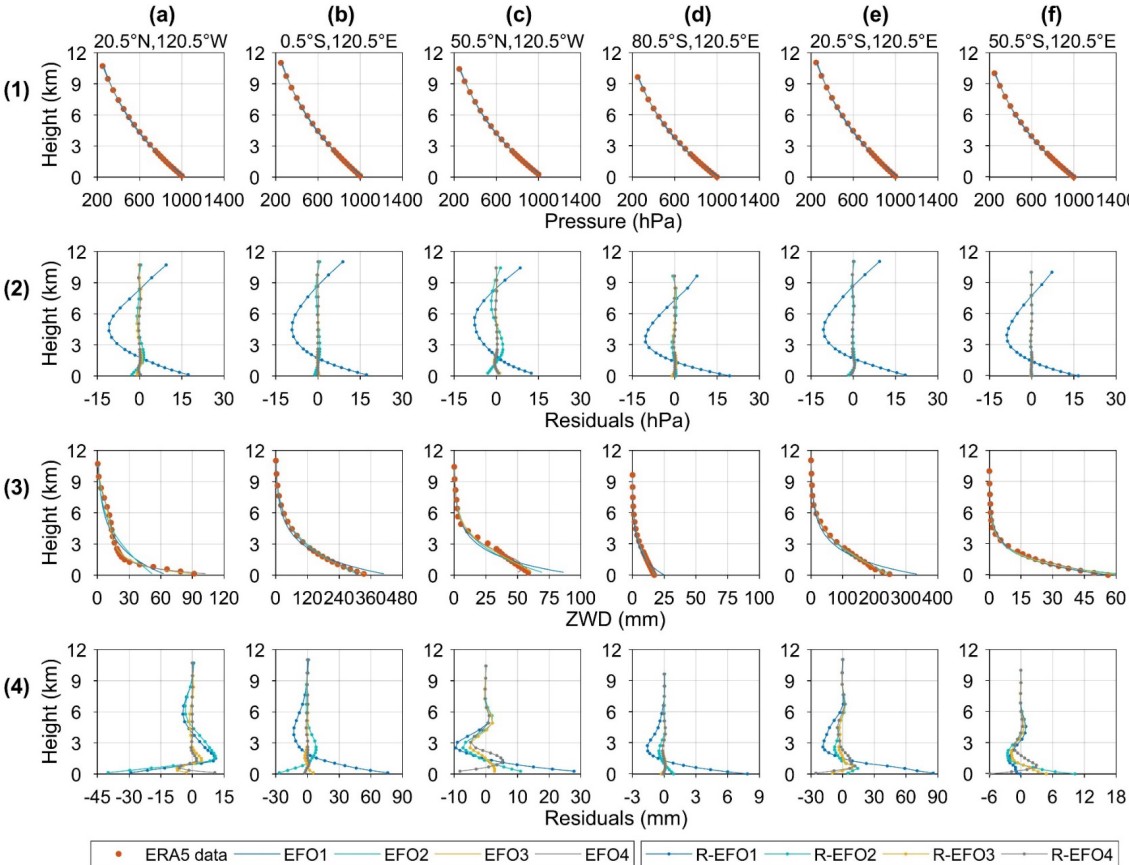

**Figure 3: The vertical data profiles (red dots), the exponential approximations and fitting residual profiles of pressure (a1-f2) and ZWD (a3-f4) at six representative ERA5 grid points. The EFO1, EFO2, EFO3 and EFO4 denote the exponential fitting function with the order of one, two, three and four, respectively. R-EOF1, R-EOF2, R-EOF3 and R-EOF4 are the corresponding fitting residuals of the four solutions.**

The global mean and maximum root mean square (RMS) values of fitting residuals obtained by four solutions are shown

in Figure 4. It is illustrated that the mean RMS of pressure fitted using EFO3 and EFO4 are less than 0.3 hPa on a global scale, they are clearly superior than those of EFO1 and EFO2. As for ZWD, the EFO2 outperforms EFO1, but the maximum RMS values still exceed 17 mm. The EFO3 generally performs identically to the EFO4, and their mean RMS values are less than 3.5 mm. As summarized above, the EFO1 and EFO2 can't reasonably account for the vertical characteristics of ZWD

and pressure. Hence, the EFO3 with relatively fewer coefficients is adopted as the core vertical correction function, which can be further expressed as:

$$\begin{cases} P_t = P_r . \exp\left[ \beta_{P1}\left( h_t - h_r \right) + \beta_{P2}\left( h_t^2 - h_r^2 \right) + \beta_{P3}\left( h_t^3 - h_r^3 \right) \right] \\ ZWD_t = ZWD_r . \exp\left[ \beta_{W1}\left( h_t - h_r \right) + \beta_{W2}\left( h_t^2 - h_r^2 \right) + \beta_{W3}\left( h_t^3 - h_r^3 \right) \right] \end{cases} \tag{13}$$

Where $P_r$ and $ZWD_r$ are the pressure and ZWD at the reference height ($h_r$), respectively. $P_t$ and $ZWD_t$ are the pressure and ZWD at the target height ($h_t$), respectively.

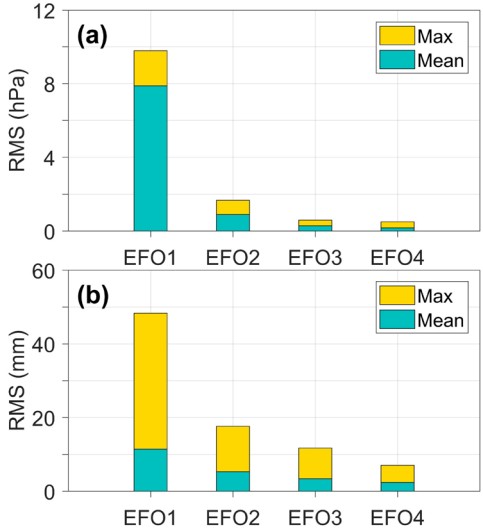

Figure 4: The global mean and maximum RMS values for pressure (a) and ZWD (b) fitted using EFO1, EFO2, EFO3 and EFO4.

### 3.3 Vertical fitting algorithm for pressure and ZWD

The Fast Fourier Transform is introduced to explore the periodicity of pressure and ZWD height scale factors. As illustrated in Figure 5, the height scale factors mainly show annual and semi-annual periods.

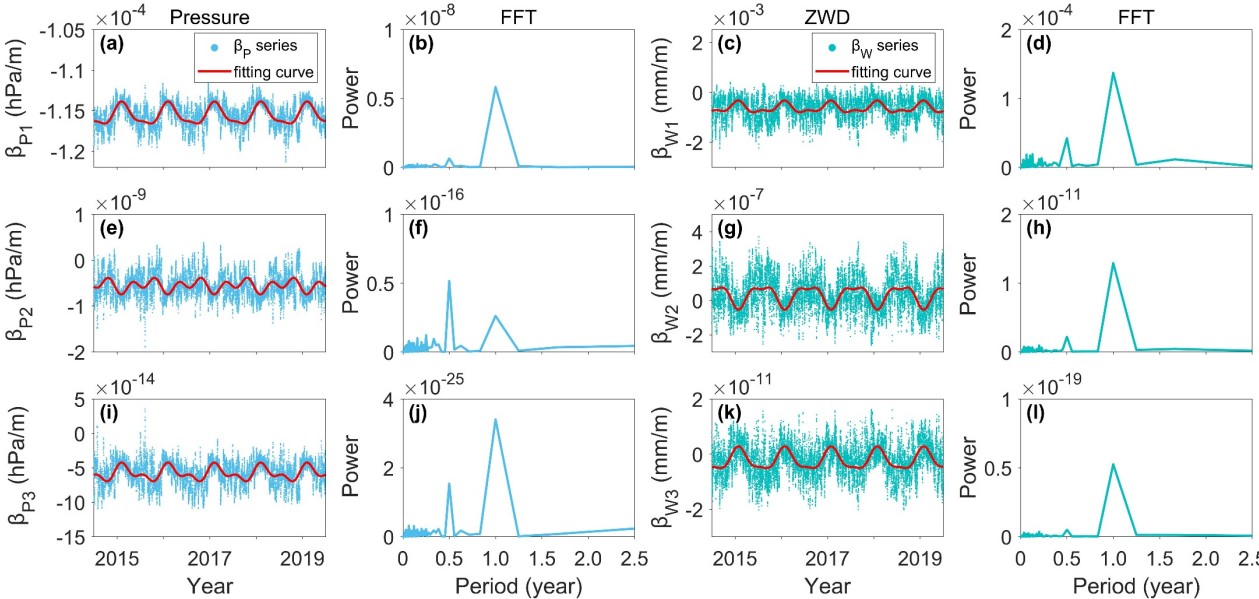

**Figure 5: The time series, fitting curves and power spectral density of $\beta_P$ and $\beta_W$ of the first (a-d), second (e-h) and third (i-l) orders at the grid point of 20.5°N, 120.5°W.**

To further investigate the spatial-temporal characteristics of pressure and ZWD height scale factors, the annual mean, annual and semi-annual amplitudes of $\beta_P$ and $\beta_W$ are surfaced as presented in Figure 6 and 7, respectively. The absolute annual mean values of $\beta_{P1}$ gradually increase from the equator to the poles, exhibiting larger negative values in the Tibet

Plateau and the Andes Mountains than the other regions in the same latitudes. The annual mean values of $\beta_{P2}$ and $\beta_{P3}$ show evident difference between ocean and land, particularly in the mid-latitudes of the western hemisphere. Additionally, large annual and semi-annual amplitudes of the three height scale factors can be found at high-latitudes. If the seasonal variations can't be properly accounted for, large errors will be introduced in the vertical extrapolation of pressure. Regarding the ZWD, the annual mean values of the three height scale factors show typical atmospheric circulation patterns, which are

characterized by the sharp gradient changes from ocean to land in the intertropical convergence zone (ITCZ). Large annual and semi-annual amplitudes of $\beta_{W1}$, $\beta_{W2}$ and $\beta_{W3}$ are observed in northern Africa and South Atlantic. Above findings demonstrate that three height scale factors of pressure and ZWD are not constant values, neither in time nor space.

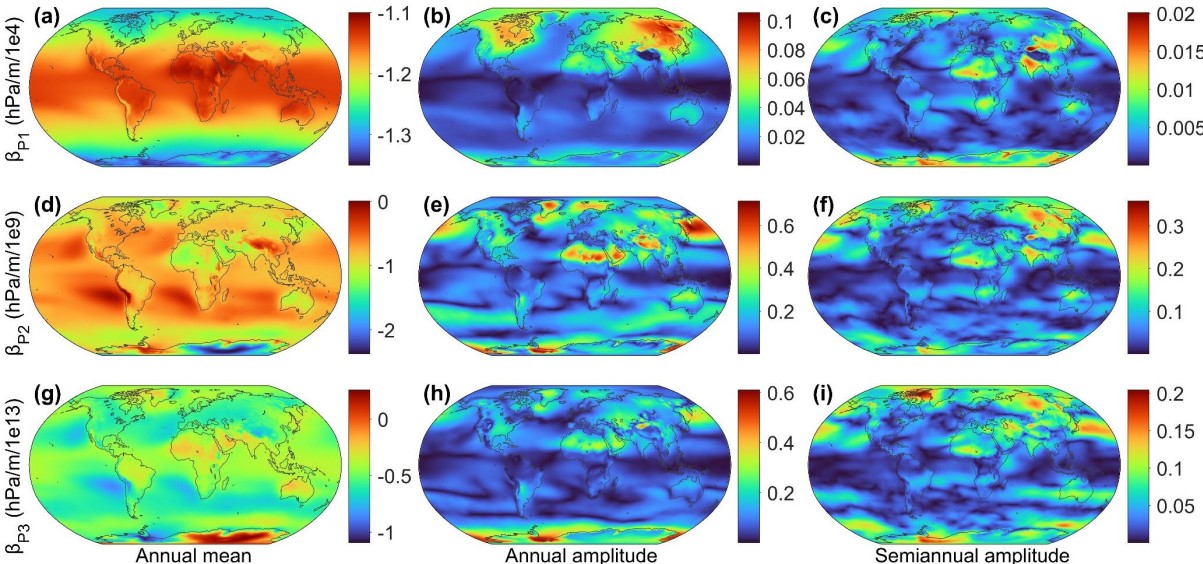

**Figure 6:** Distributions of the annual mean values, annual and semi-annual amplitudes of $\beta_{P1}$ (a-c), $\beta_{P2}$ (d-f) and $\beta_{P3}$ (g-i). Note that the colorbar scales of each subgraph are different.

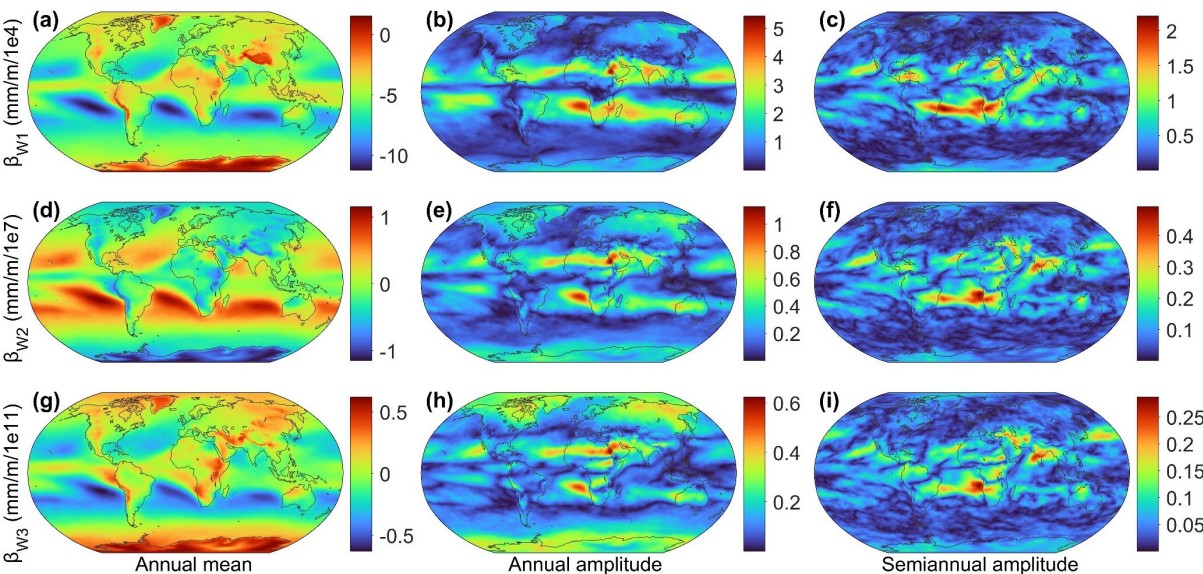

**Figure 7:** Distributions of the annual mean values, annual and semi-annual amplitude of $\beta_{W1}$ (a-c), $\beta_{W2}$ (d-f) and $\beta_{W3}$ (g-i). Note that the colorbar scales of each subgraph are different.

To enhance the vertical performance of the new model, the spatial grid windows with the same horizontal resolution (1°×1°) as the surface model in section 3.1 are adopted to characterize the horizontal spatial variations of height scale factors. Meanwhile, the following harmonic functions are used to fit the $\beta_P$ and $\beta_W$ time series at each grid point:

$$\beta_{Pi}(\beta_{Wi}) = A_0^i + \sum_{n=1}^{2}[A_n^i \sin(\frac{2\pi n.\text{DOY}}{365.25}) + B_n^i \cos(\frac{2\pi n.\text{DOY}}{365.25})], i = 1,2,3 \tag{14}$$

Where $A_0^i$, $A_n^i$ and $B_n^i$ are annual mean, annual and semi-annual amplitudes of the $n$th order height scale factors of pressure or ZWD.

By integrating equation (14) with (15), the final vertical expression of pressure and ZWD are derived as follow:

$$P_t(ZWD_t) = P_r(ZWD_r).\exp\left\{\sum_{i=1}^{3}\left[A_0^i + \sum_{n=1}^{2}\left[A_n^i \sin\left(\frac{2\pi n.\text{DOY}}{365.25}\right) + B_n^i \cos\left(\frac{2\pi n.\text{DOY}}{365.25}\right)\right].\left[(H_t)^i - (H_r)^i\right]\right]\right\} \tag{15}$$

Finally, combining the surface (5) and vertical correction (16) modules, the improved global pressure and ZWD (IGPZWD) model is expressed as follow:

$$R^{IGPZWD} = \left\{a_0 + \sum_{l=1}^{2}\left[a_l \sin\left(\frac{2\pi l.\text{DOY}}{365.25}\right) + b_l \cos\left(\frac{2\pi l.\text{DOY}}{365.25}\right)\right] + \sum_{l=1}^{2}\left[c_l \sin\left(\frac{2\pi l.\text{HOD}}{24}\right) + d_l \cos\left(\frac{2\pi l.\text{HOD}}{24}\right)\right]\right\} \times$$
$$\exp\left\{\sum_{i=1}^{3}\left[A_0^i + \sum_{n=1}^{2}\left[A_n^i \sin\left(\frac{2\pi n.\text{DOY}}{365.25}\right) + B_n^i \cos\left(\frac{2\pi n.\text{DOY}}{365.25}\right)\right].\left[(H_t)^i - (H_r)^i\right]\right]\right\} \tag{16}$$

The development and use of the IGPZWD model are summarized in the flowchart depicted in Figure 8, including surface and vertical correction modules. With the geodetic location and time specified as DOY and HOD as inputs, the pressure and ZWD of the nearest four model grid points at the target height are determined according to equation (17). Thereafter, a bilinear interpolation method is carried out to calculate the target pressure and ZWD. Furthermore, the target ZHD and ZTD are obtained based on the Saastamoinen model as follows:

$$ZHD_s = \frac{0.0022768.P^{IGPZWD}}{1 - 0.00266\cos 2\varphi - 0.00028H_s} \tag{17}$$

$$ZTD_s = ZHD_s + ZWD^{IGPZWD} \tag{18}$$

Where $P^{IGPZWD}$ and $ZWD^{IGPZWD}$ are the pressure and ZTD predicted by the IGPZWD model, respectively.

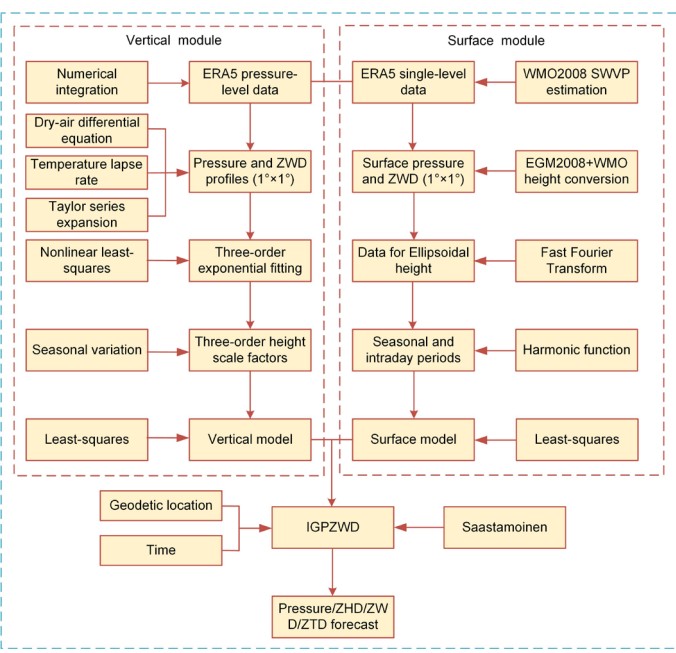

**Figure 8: Flowchart depicting the development and use of the IGPZWD model.**

## 4 Validation and discussion

In this section, the accuracy and spatial stability of the IGPZWD model are comprehensively investigated and analyzed using the ERA5 hourly pressure-level data and radiosonde data profiles below 15 km in 2020. In addition to the most commonly used GPT3 model, the state-of-the-art IGPT model and GTrop model are introduced to verify the accuracy advantages of the

245 pressure and ZWD predicted by the IGPZWD model, respectively. Furthermore, the performance of IGPZWD-predicted ZTD is evaluated by comparing with GPT3, GTrop and reconstructed GPT3 models (RGPT3).

### 4.1 Evaluation with ERA5-derived pressure and ZWD

It is noted that GTrop model don't directly provide pressure prediction. Although the pressure can be converted from GTrop-predicted ZHD based on the Saastamoinen model, it will result in a non-negligible loss of accuracy. Consequently, taking the

250 pressure and ZWD profiles from ERA5 in 2020 as reference, the global accuracies of those predicted by the GPT3, IGPT and IGPZWD models at four representative pressure levels are presented in Figure 9. The GPT3 model generally overestimates the pressure at four levels, showing systematic positive biases which gradually increase with altitude, and the mean RMS at 350 hPa even exceeds 28 hPa. The IGPT-predicted pressure shows greater consistency with the ERA5 data than that of GPT3 model at the upper three levels, but it performs poorly in the bottom level. The reason is that the IGPT3

model applies the same inaccurate pressure extrapolation method as the GPT3 model below 2 km (Li et al., 2021). Compared

to the IGPT and GPT3 models, the IGPZWD model shows better performance at each level and achieves overall unbiased pressure prediction in the tropical regions.

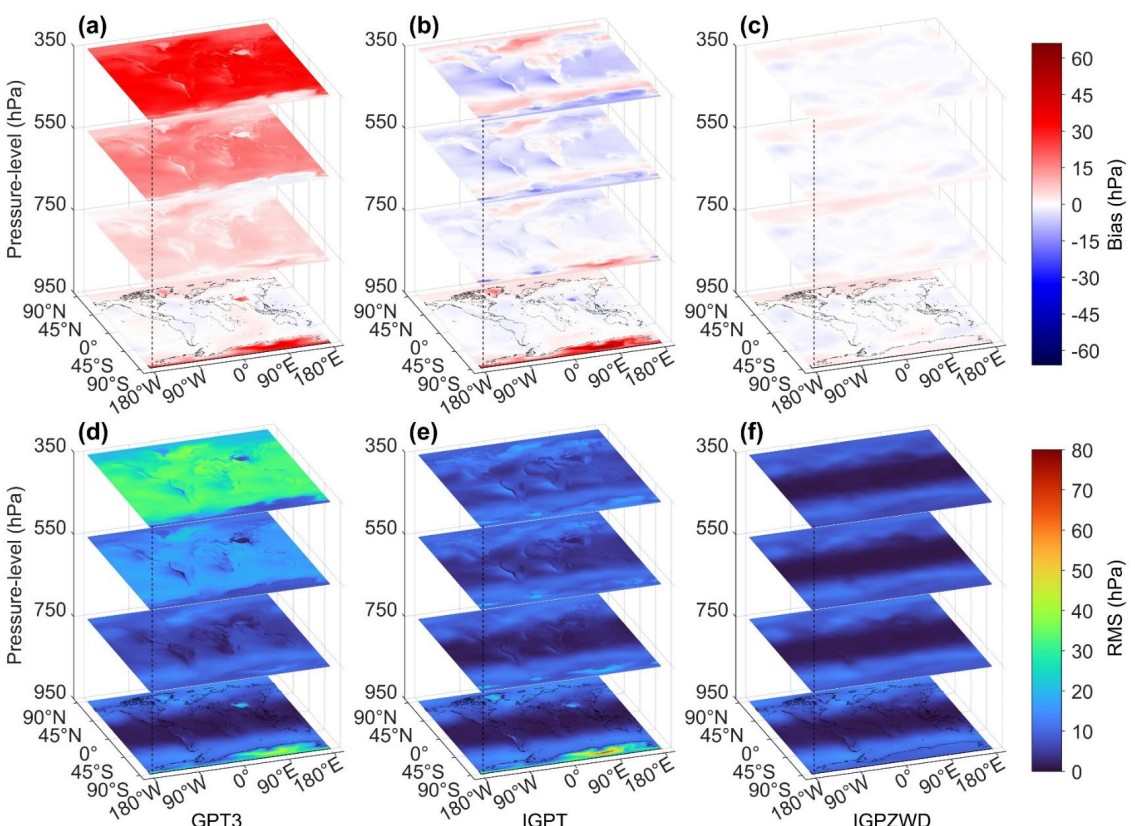

**Figure 9: Global distribution of bias (a-c) and RMS (d-f) for the pressure predicted by the GPT3, IGPT, and IGPZWD models validated using the ERA5 pressure at the levels of 950, 750, 550 and 350 hPa in 2020.**

Figure 10 depicts the vertical accuracies of pressure profiles predicted by GTP3, GTrop and IGPZWD models in three representative regions with different climatic environments and geographical locations. IGPZWD model exhibits overall optimal accuracy and stability with no significant sudden change. In the Tibet Plateau and Antarctic, the RMS and bias values of GPT3 model show evident and sharp trends of first decreasing and then increasing with altitude due to unreasonable pressure extrapolation method. Above 800 hPa, IGPT model tends to underestimate the pressure in the Andes mountains region, inducing systematic negative bias and relatively poorer RMS. Overall, the IGPZWD model achieves great pressure prediction on both the surface and the upper air, which benefits from the consideration of the seasonal variations for the pressure height scale factors.

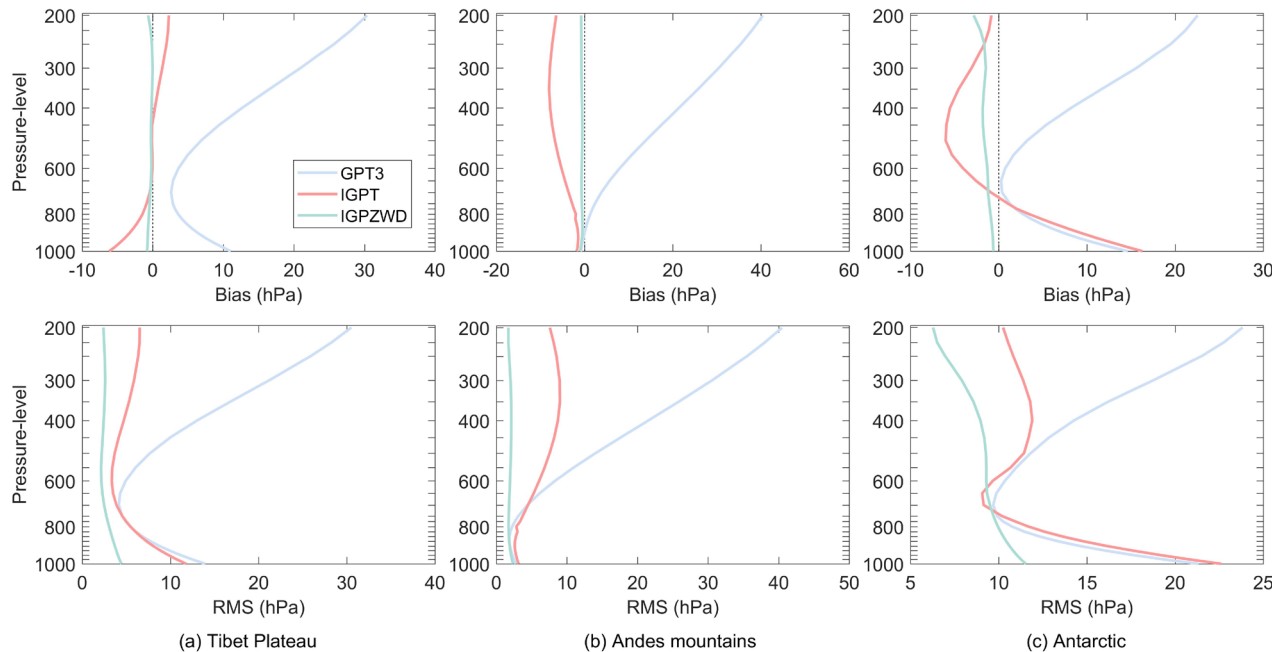

 **Figure 10: Bias and RMS of pressure profiles predicted by the GPT3, IGPT, and IGPZWD models validated using the ERA5 pressure from 1000 to 200 hPa in 2020. The three selected regions are Tibet Plateau (a), Andes mountains (b) and Antarctica (c).**

The statistical results of model-predicted ZWD validated using ERA5 profiles are shown in Figures 11 and 12. The magnitude of ZWD gradually decreases with increasing altitude, but the GPT3 model still shows a significant systematic positive bias at 350 hPa. This may be due to inaccurate estimation of the water vapor decrease factor, resulting in the accumulation of vertical errors. In contrast to GPT3 model, the GTrop and IGPZWD perform better at 550 and 350 hPa, showing smaller bias and RMS values in low latitudes. Furthermore, high-frequency moist convection effect is generally accompanied by drastic spatial-temporal changes of water vapor in the tropics, which makes it difficult to capture the temporal variation of ZWD using empirical models. Correspondingly, the ZWDs predicted by the three models show poor consistencies with ERA5 ZWD in the low-latitude oceans at 950 hPa. Nonetheless, the IGPZWD model exhibits the smallest mean bias at 350 hPa, with a mean RMS of 0.9 mm which is corresponding to 63.4% and 29.5% improvements against GPT3 and GTrop models, respectively.

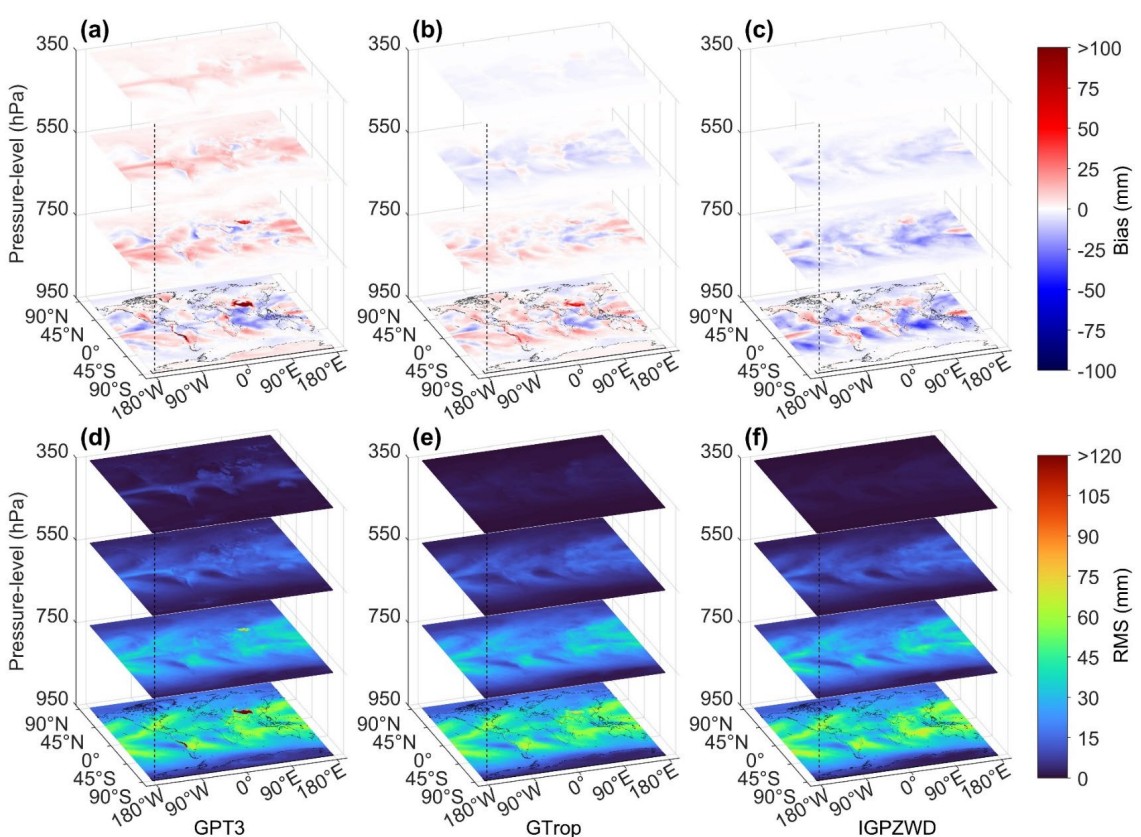

**Figure 11: Global distribution of bias (a-c) and RMS (d-f) for the ZWD predicted by the GPT3, GTrop, and IGPZWD models validated using the ERA5 ZWD at the levels of 950, 750, 550 and 350 hPa in 2020.**

Figure 12 illustrates that the GPT3 and GTrop models exhibit obviously positive bias in the Andes Mountains and Tibet Plateau below 800 hPa, and the RMS values of GPT3 exceeds 100 mm in the Tibetan Plateau region. In contrast, the IGPZWD model exhibits smaller bias values in these regions, and the RMS values are less than 40 mm. In the Antarctica, IGPZWD outperform all the other two models, achieving overall unbiased ZWD prediction above 400 hPa. It is concluded that IGPZWD model-predicted ZWD has a certain vertical accuracy advantage compared to GTrop and it is significantly

more accurate than GPT3. Although IGPZWD-predicted ZWD exhibit superior performance in high-altitude areas, the improvement in surface is negligible. It is concluded that developing surface ZWD models is challenging. Nevertheless, substantial studies have proven that the cubic polynomial can effectively improve the fitting effect of ZWD profiles at low altitudes, which can be the algorithm reference for future enhanced model construction (Li et al., 2023; Xu et al., 2023).

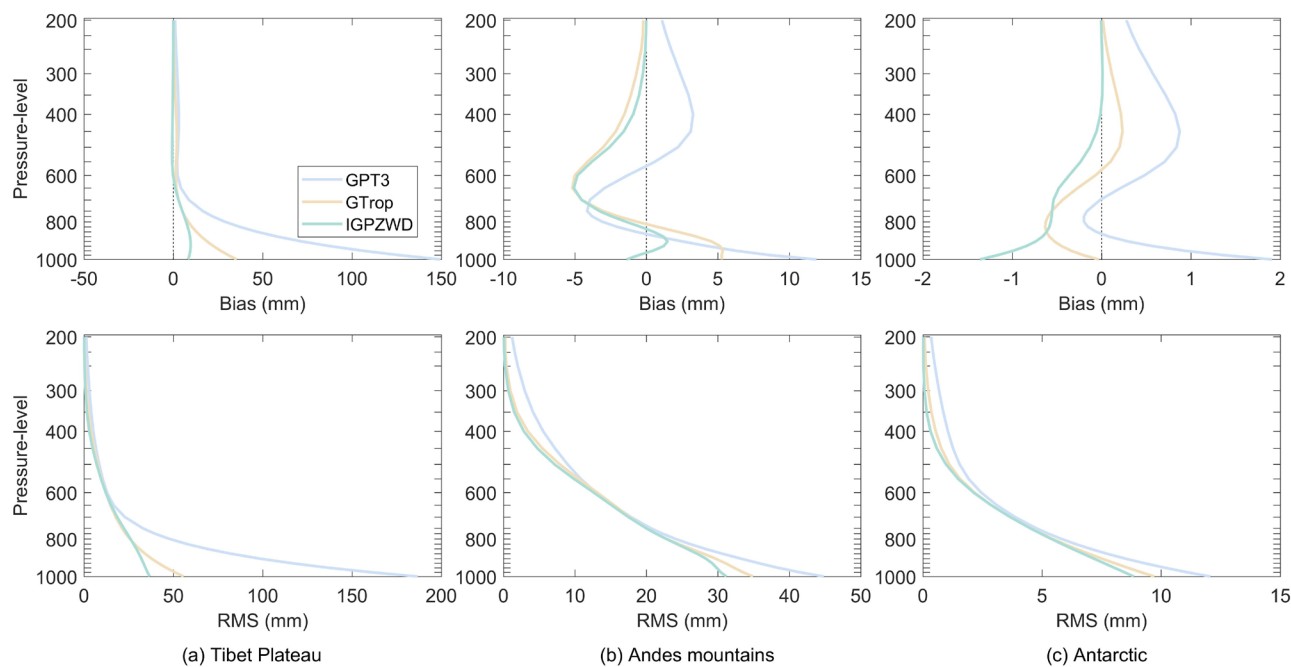

**Figure 12: Bias and RMS of ZWD profiles predicted by the GPT3, IGPT, and IGPZWD models validated using the ERA5 ZWD from 1000 to 200 hPa in 2020. The three selected regions are Tibet Plateau (a), Andes mountains (b) and Antarctica (c).**

### 4.2 Validation with radiosonde-derived pressure and ZWD

Initially, different height systems between radiosonde and model have been unified according to the Earth Gravitational Model (EGM) 2008 model (Pavlis et al., 2012) and World Meteorological Organization (WMO) 2018 standard measurement (Yuan et al., 2023). Subsequently, taking the longitude, latitude, heights, DOY and UTC of each data point on the filtered radiosonde profiles below 15 km as inputs, four models are employed to predict the corresponding ZWD and atmospheric pressure. Thereafter, the pressure and ZWD profiles derived from radiosonde observation in 2020 are used as references to

evaluate the model-predicted pressure and ZWD. To investigate the applicability of the three models at different height ranges below 15 km, the accuracies are statistically analyzed with a vertical sampling interval of 3 km.

     The bias and RMS values of pressure predicted by GPT3, IGPT and IGPZWD models at three Temperature zones are presented in figure 13. It can be seen that the GPT3 model exhibits a systematic positive bias above 3 km, with a large mean bias value of 29 hPa in the temperate zone at the range of 12-15 km. Evidently, the accuracy of the GPT3 model gradually

decreases with the increase of altitude, indicating that its pressure extrapolation scheme is inapplicable when the height difference is large. The IGPT model exhibits superior accuracy than the GPT3 model in the temperate and tropical regions where the intraday variations of pressure are strong, which benefits from the consideration of diurnal and semi-diurnal terms in pressure. IGPZWD model further effectively improves the accuracy compared to IGPT model, achieving almost unbiased estimation of pressure with RMS improvements of 21.8-41.1% in tropical and 68.7-82.9% in temperate zones. In addition,

the RMS of IGPZWD model has improved by over 94% compared to GPT3 model beyond 6 km in tropical regions, indicating the feasibility of the proposed vertical correction algorithm. In summary, it has been verified that exponential function with three orders has stronger accuracy advantages and robustness in vertical pressure extrapolation compared to existing models such as virtual temperature and the adiabatic models. In addition, we further achieved accurate prediction of the height scale factor time series for pressure.

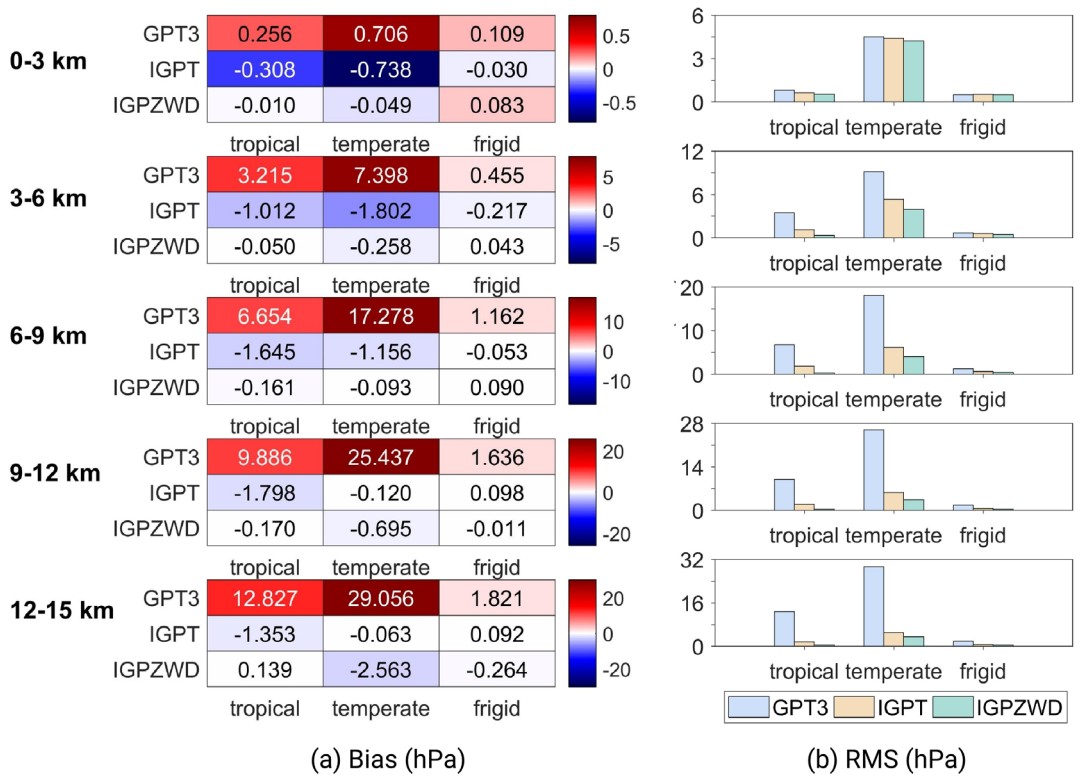

(a) Bias (hPa)  (b) RMS (hPa)

**Figure 13: Mean bias (a) and RMS (b) values (d1-d5) for pressure predicted by the GPT3, IGPT and IGPZWD models validated using the radiosonde pressure data at five height ranges of the tropical, temperate, and frigid zones.**

Table 1 summarizes the mean pressure bias and RMS values of each height range. The RMS values of IGPZWD do not exceed 5.3 hPa at five height ranges, showing a significant accuracy advantage compared to GPT3, with an improvement of
325 up to 90% for 12-15 km. In contrast to the IGPT model, the IGPZWD model exhibits smaller negative bias values and further improves the performance beyond 3 km with RMS improvements of 32.4-51.8%, indicating the feasibility of the proposed vertical correction algorithm. The magnitude of ZWD in high altitude is small, and thus the pressure is the main factor restricting the accuracy of ZTD according to the rule of uncertainty propagation. It is implied that IGPZWD may provide superior prior tropospheric constraints for GNSS positioning of high-altitude platforms.

**Table 1: Mean bias and RMS values for pressure predicted by the GPT3, IGPT and IGPZWD models at five height ranges.**

| Height (km) | Bias (hPa) | | | RMS (hPa) | | |
|---|---|---|---|---|---|---|
| | GPT3 | IGPT | IGPZWD | GPT3 | IGPT | IGPZWD |
| 0-3 | 1.1 | -1.1 | 0.0 | 5.8 | 5.6 | 5.3 |
| 3-6 | 11.1 | -3.0 | -0.3 | 13.3 | 7.1 | 4.8 |
| 6-9 | 25.1 | -2.9 | -0.2 | 26.2 | 8.6 | 4.8 |
| 9-12 | 37.0 | -1.8 | -0.9 | 37.5 | 8.3 | 4.0 |
| 12-15 | 43.7 | -1.3 | -2.7 | 44.0 | 7.0 | 4.2 |

The bias and RMS values of pressure predicted by GPT3, IGPT and IGPZWD models at three Temperature zones are presented in figure 14. Significant negative bias values of the three models are observed in Southeast Asia below 3 km, which is attributed to the local strong annual and semi-annual amplitudes of ZWD. The GPT3 model exhibits generally positive bias values and large RMS values above 3 km in tropical and temperate zones, which again demonstrate that it can't provide reliable ZWD information in high-altitude areas. Although the GTrop model shows slight accuracy advantage below 3 km, while it performs worse than the IGPZWD model above 3 km. Compared to the GTrop model, IGPZWD model achieves RMS improvements of 14.5-27.8% and 10.6-48.5 % beyond 6 km in temperate and tropical zones, respectively. And the order of magnitude of improvement increases with height, confirming the advantages of high-order exponents in simulating ZWD profiles. Previous scholars have used piecewise and high-order functions to characterize the vertical nonlinear variation of ZWD (Hu and Yao, 2019; Zhu et al., 2022). We have also verified the feasibility of this kind of algorithm using 1-year radiosonde data. Moreover, it has been proven that considering time-varying height scale factor can further improve the long-term forecasting accuracy.

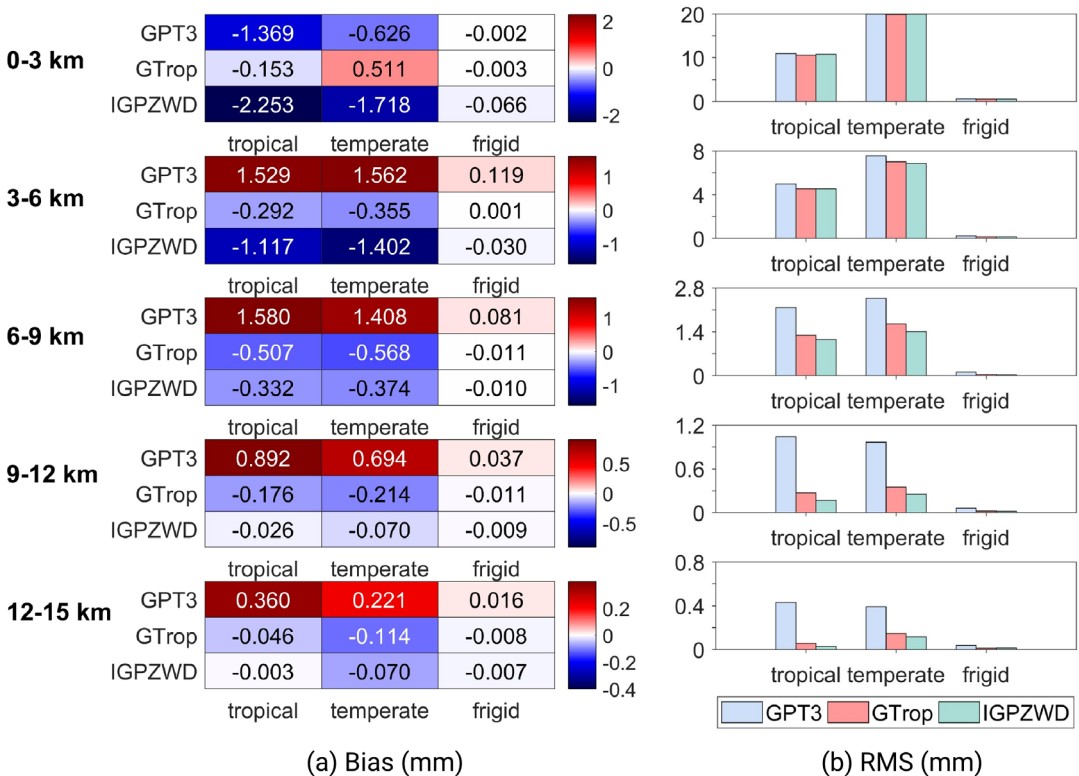

(a) Bias (mm)  (b) RMS (mm)

**Figure 14: Mean bias (a) and RMS (b) values (d1-d5) for pressure predicted by the GPT3, IGPT and IGPZWD models validated using the radiosonde ZWD data at five height ranges of the tropical, temperate, and frigid zones.**

Table 2 summarizes the mean bias and RMS values corresponding to each height range. The mean RMS values of IGPZWD model are less than 2.6 mm beyond 6 km, showing the improvements ranges from 45.8% to 81.4% over the GPT3 model and from 13.3% to 31.3% over the GTrop model respectively. The above results indicate that the IGPZWD model achieves optimal vertical accuracy and stability on a global scale.

**Table 2: Mean bias and RMS values for ZWD predicted by the GPT3, IGPT, and IGPZWD models at five height ranges.**

| Height (km) | Bias (mm) | | | RMS (mm) | | |
|---|---|---|---|---|---|---|
| | GPT3 | GTrop | IGPZWD | GPT3 | GTrop | IGPZWD |
| 0-3 | -2.0 | 0.4 | -4.0 | 31.6 | 31.1 | 31.4 |
| 3-6 | 3.2 | -0.6 | -2.5 | 12.8 | 11.7 | 11.5 |
| 6-9 | 3.1 | -1.1 | -0.7 | 4.8 | 3.0 | 2.6 |
| 9-12 | 1.62 | -0.40 | -0.11 | 2.07 | 0.64 | 0.44 |
| 12-15 | 0.60 | -0.17 | -0.08 | 0.86 | 0.21 | 0.16 |

### 4.3 Validation with radiosonde-derived ZTD

The vertical correction for pressure in the GPT3 model is realized by the following method:

$$\begin{cases} T_v = T.(1+0.6067.Q) \\ P_t = P_r.\exp\left(-\frac{g_m.dMtr}{Rg.T_v}.\Delta h\right) \end{cases}$$

(19)

Where $T$, $Q$ and $T_v$ are the temperature, specific humidity and virtual temperature at the reference height, respectively. $dMtr$, $Rg$ and $\Delta h$ denote the molar mass of dry-air, universal gas constant and the corrected height difference, respectively. Equation (19) is essentially based on the assumption of isothermal atmosphere, but the actual atmospheric state does not meet the condition, except for the tropopause. Significant errors will be introduced when using the isothermal model to carry out pressure extrapolation of large height difference, resulting in poor accuracy of ZTD. Therefore, to enhance the comparability of GPT3 ZTD in the high-altitude areas and investigate the potential applicability of the proposed vertical correction method, the pressure extrapolation module of the GPT3 model is replaced by that of IGPZWD model. The calculation method of ZTD for the reconstructed GPT3 (RGPT3) model is as follow:

$$ZTD_{RGPT3} = \frac{0.0022768.P_{GPT3}.\exp\left\{\sum_{i=1}^{3}\left[\beta_{Pi}.\left[(H_s)^i - (H_r)^i\right]\right]\right\}}{1 - 0.00266\cos 2\varphi - 0.00028H_s} + 10^{-6}(k_2' + k_3/T_m)\frac{R_d}{(\lambda+1)g_m}e_s$$

(20)

Where $P_{GPT3}$ is the pressure at the reference height, $k_2'$ and $k_3$ denote the empirical coefficients. $e_s$ and $\lambda$ are the water vapor pressure and corresponding decrease factor, respectively. The accuracies of the GPT3, RGPT3, GTrop and IGPZWD models are evaluated with respect to the radiosonde-derived ZTD below 15 km in 2020. Figure 15 depicts the RMS improvement of RGPT3 ZTD compared to GPT3 ZTD at each station and five height ranges. The RGPT3 model achieves a comprehensive accuracy improvement, with RMS improvements over 60% at most stations. The improvements range from 3.4% to 88.4% under 15 km, implying the feasibility and wide applicability of the new pressure vertical correction method.

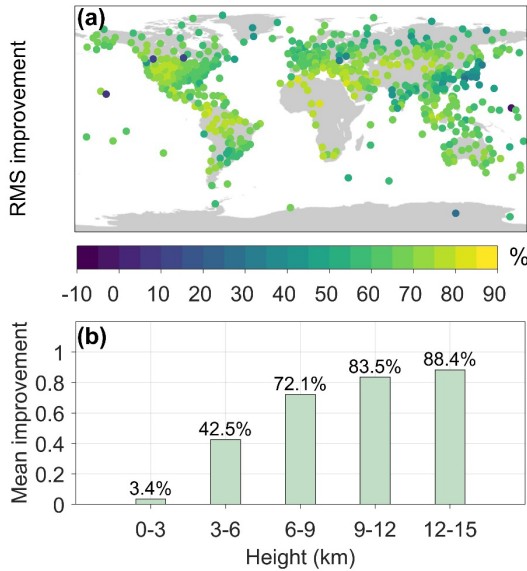

**Figure 15: RMS improvement of RGPT3 ZTD compared to GPT3 ZTD at 565 stations (a) and the height ranges of 0-3, 3-6 6-9, 9-12 and 12-15 km (b).**

Table 3: The maximum, minimum, and mean values of bias and RMS for ZTD predicted by the GPT3, RGPT3, GTrop and IGPZWD models.

| Model | Bias (mm) | | | RMS (mm) | | |
|---|---|---|---|---|---|---|
| | Min | Max | Mean | Min | Max | Mean |
| GPT3 | -1.4 | 79.3 | 48.2 | 23.1 | 104.3 | 65.4 |
| RGPT3 | -48.1 | 16.4 | 0.3 | 9.9 | 100.2 | 23.0 |
| GTrop | -55.0 | 14.6 | -1.9 | 10.4 | 103.9 | 24.3 |
| IGPZWD | -52.2 | 13.8 | -0.8 | 9.5 | 104.0 | 22.4 |

The global accuracies of ZTD predicted by the GPT3, RGPT3, GTrop and IGPZWD models are shown in Table 3 and Figure 15. As summarized in Table 3, the mean RMS values of IGPZWD is 22.4 mm, which is corresponding to 65.7%, 2.4% and 7.8% improvements against the GPT3, RGPT3 and GTrop models respectively. As illustrated in Figure 16, the GPT3 model exhibits significant positive bias values caused by inaccurate pressure estimation, with RMS values over 60 mm at most stations in low and middle latitudes. In contrast to the GPT3 model, the RGPT3, GTrop and IGPZWD models achieve overall unbiased ZTD estimations, and the proportions of RMS values below 30 mm for the three models account for 92.4%, 87.8% and 92.9% respectively. Moreover, Figure 17 depicts the mean accuracies of ZTD predicted by the four models at five height ranges. The optimized RGPT3 model outperforms GTrop model beyond 6 km, further enhancing the vertical applicability of the GPT3 model. Overall, the IGPZWD model exhibits optimal accuracy at all height ranges, which is attributed to the comprehensive consideration of periodic terms and optimized vertical correction algorithm in terms of pressure and ZWD.

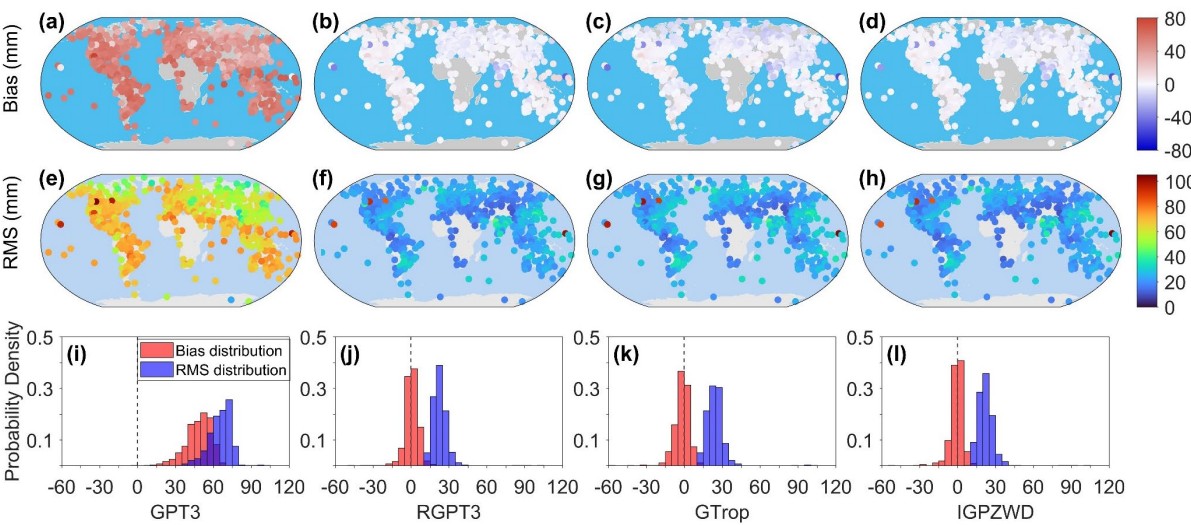

Figure 16: Global distribution of bias (a-d), RMS (e-h) and the corresponding probability density histograms (i-l) for the ZTD predicted by the GPT3, RGPT3, GTrop and IGPZWD with respect to the radiosonde-derived ZTD.

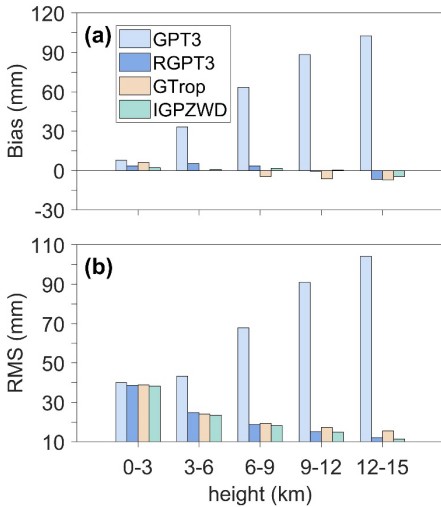

Figure 17: Overall bias (a) and RMS (b) values for ZTD predicted by the GPT3, RGPT3, IGPT and IGPZWD models at the height ranges of 0-3, 3-6, 6-9, 9-12 and 12-15 km.

To sum up, the proposed IGPZWD model can provide high-quality tropospheric parameters prediction below 15 km on a global scale. The IGPZWD model will be of great significance for the tropospheric augmentation in real-time GNSS positioning, and it has broad application prospects in real-time water vapor sounding and extreme weather forecasting. Despite the complexity of the inversion process for tropospheric parameters, the overall performance of IGPZWD model is encouraging. In the future, the model coefficients will be further simplified to balance the computation efficiency and accuracy.

**5 Conclusions**

Accurate atmospheric pressure and ZWD information are crucial for real-time GNSS precise positioning and meteorological applications. With 5-year ERA5 hourly data, we reveal the spatial-temporal characteristics of pressure and ZWD and

405 propose an empirical global pressure and ZWD grid model with broader operating space, called IGPZWD, which incorporates the diurnal and semi-diurnal harmonics. The optimal exponential function with three orders is adopted as the core vertical fitting scheme, and the seasonal variations of height scale factors are taken into consideration to further optimize the vertical accuracy. Consequently, the IGPZWD model can quickly provide accurate pressure, ZWD, ZHD and ZTD estimates for any selected time and location over globe.

The performance of IGPZWD is evaluated with the ERA5 and radiosonde profiles data in 2020. Taking the ERA5 pressure profiles as reference, the IGPZWD model outperform the GPT3 and IGPT models, achieving overall unbiased pressure prediction in the tropical regions and significant RMS improvement in the Antarctic, Tibet Plateau and Andes mountains on both the surface and the upper air. Regarding the ZWD, the IGPZWD model exhibits greater consistency with ERA5 ZWD than the GPT3 and GTrop models at higher levels, achieving optimal accuracy over the globe and overall

unbiased ZWD prediction in the Antarctic. The validation based on radiosonde profiles data indicate that the pressure predicted by the IGPZWD model show better performance than that of GPT3 and IGPT models. The pressure accuracy of the IGPZWD model is improved by 32.4-51.8% compared to that of IGPT model beyond 3 km. Above 6 km, the RMS of ZWD predicted by the IGPZWD model is improved by 45.8-81.4% and 13.3%-31.3% in contrast to that of GPT3 and GTrop models respectively. Furthermore, the mean RMS value of ZTD predicted by IGPZWD is 22.4 mm, which achieves 65.7%,

2.4% and 7.8% improvements against that of GPT3, RGPT3 and GTrop models respectively.

In summary, the proposed optimized vertical correction algorithm weakens the cumulative error caused by large correction height difference, which effectively improves the accuracy and stability of pressure, ZWD and ZTD in high-altitude areas.

## 6 Appendix A

To reveal the applicability of the three models at different height ranges below 15 km, the accuracies of model-predicted pressure and ZWD profiles are statistically validated using radiosonde profiles data with a vertical sampling interval of 3 km. The bias and RMS values of pressure are presented in Figure A1 and A2, respectively, while those of ZWD are shown in Figures A3 and A4. The following figures can effectively demonstrate the global and regional accuracy advantage of IGPZWD model in different height ranges, providing indicators and references for users in different regions.

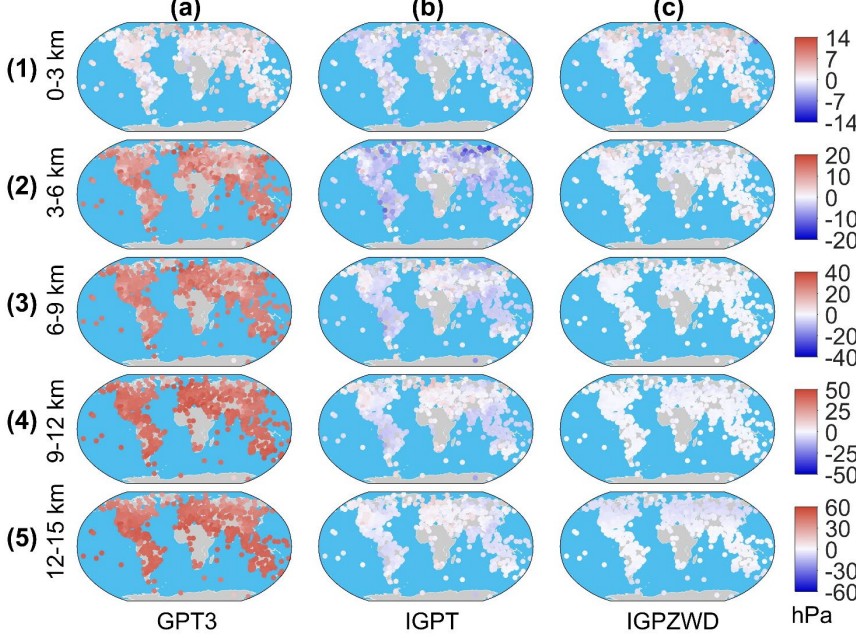

**Figure A1: Distribution of bias for pressure predicted by the GPT3, IGPT and IGPZWD models validated using the radiosonde pressure data at the height ranges of 0-3 (a1-c1), 3-6 (a2-c2), 6-9 (a3-c3), 9-12 (a4-c4) and 12-15 (a5-c5) km in 2020.**

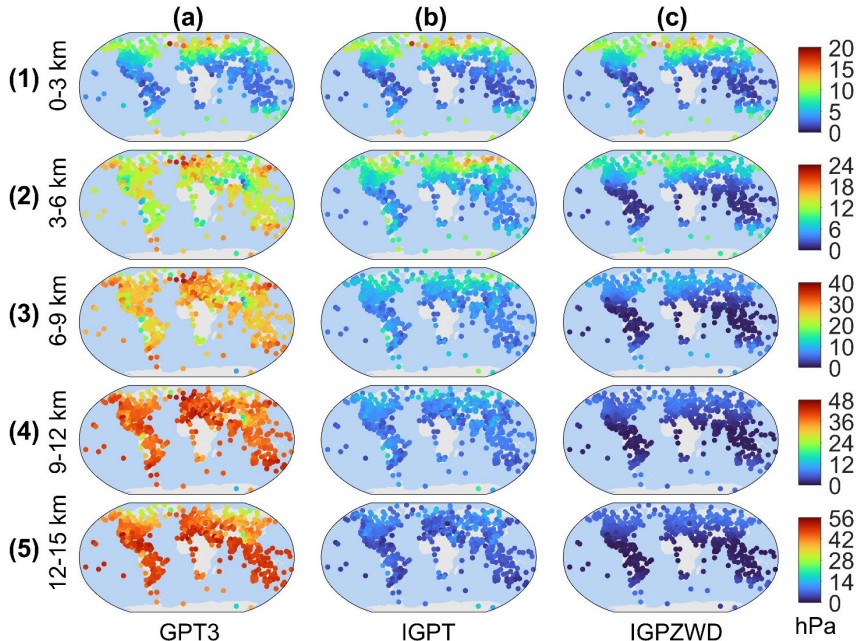

**Figure A2: Distribution of RMS for pressure predicted by the GPT3, IGPT and IGPZWD models validated using the radiosonde**
**pressure data at the height ranges of 0-3 (a1-c1), 3-6 (a2-c2), 6-9 (a3-c3), 9-12 (a4-c4) and 12-15 (a5-c5) km in 2020.**

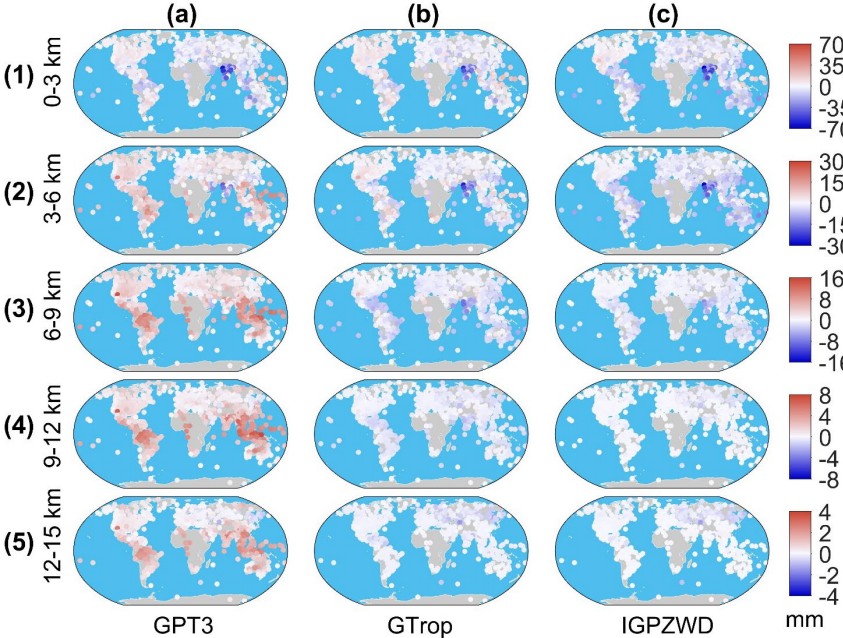

**Figure A3: Distribution of bias for ZWD predicted by the GPT3, GTrop and IGPZWD models validated using the radiosonde-derived ZWD at the height ranges of 0-3 (a1-c1), 3-6 (a2-c2), 6-9 (a3-c3), 9-12 (a4-c4) and 12-15 (a5-c5) km in 2020.**

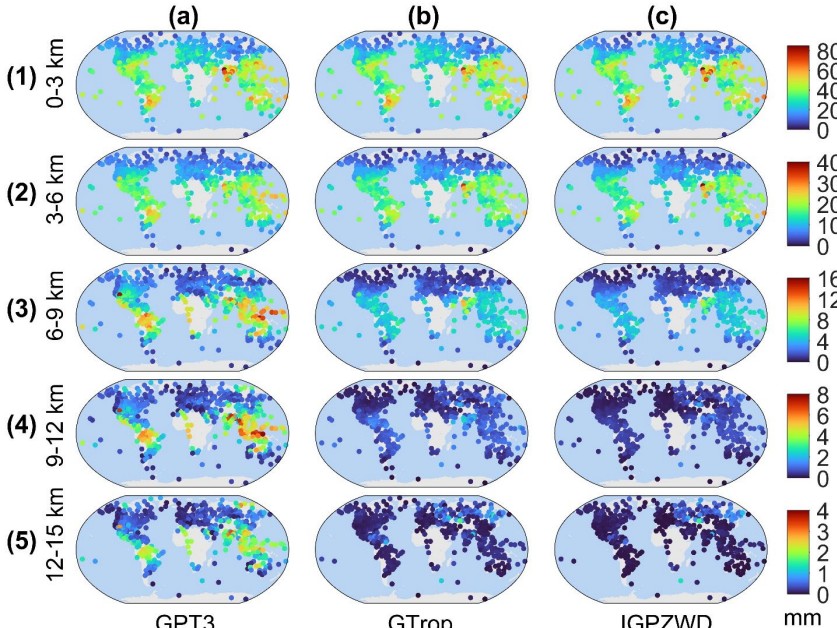

**Figure A4: Distribution of bias for ZWD predicted by the GPT3, GTrop and IGPZWD models validated using the radiosonde-derived ZWD at the height ranges of 0-3 (a1-c1), 3-6 (a2-c2), 6-9 (a3-c3), 9-12 (a4-c4) and 12-15 (a5-c5) km in 2020.**

*Code and data availability*. The open-source codes and coefficient matrix files of the IGPZWD model are available at https://github.com/LNTUgx/GNSS/tree/main/IGPZWD_model or https://doi.org/10.5281/zenodo.10574193. ERA5 reanalysis products can be obtained from https://www.ecmwf.int/en/forecasts/datasets/reanalysis-datasets/era5. Radiosonde can be downloaded from https://www.ncei.noaa.gov/products/weather-balloon/integrated-global-radiosonde-archive.

*Author contributions*. CJ designed the experimental processes and core algorithm, reviewed and optimized the paper, and provided fundings. XG conducted the code implementation for data processing and visualization, and wrote the original draft. HZ adjusted the structure of the paper and provided funding. SW verified the feasibility of the algorithm. SL provided scripts for obtaining basic data. SC and GL conducted the correction and verification of basic data.

*Competing interests.* The contact author has declared that none of the authors has any competing interests.

*Acknowledgements*. The authors would like to thank the three international research teams for providing open-source codes in terms of GPT3, IGPT and GTrop. The Integrated Global Radiosonde Archive and Copernicus Climate Change Service are appreciated for providing radiosonde data and ERA5 reanalysis products. We also thank the supercomputing system in the Supercomputing Center, Shandong University, Weihai.

*Financial support*. This research is founded by the National Natural Science Foundation of China (42030109), the Scientific Study Project for lnstitutes of Higher Learning, Ministry of Education, Liaoning Province (LJKMZ20220673), the Project

supported by the State Key Laboratory of Geodesy and Earths' Dynamics, Innovation Academy for Precision Measurement Science and Technology (SKLGED2023-3-2).

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
