# Peer review of "An improved global pressure and ZWD model with optimized vertical correction considering the spatial-temporal variability of multiple height scale factors"

_EGUsphere, 2023_

## Author Comment (AC3)

**Response to Referee 1 Comments**

General comments:

In the paper the authors describe a global data base of pressure and zenith wet delay (ZWD) to be used as a priori information in new estimates of ZWD. The authors show that the performance is better than the current existing models which means that this is an important and interesting contribution to the field. The paper however, is hard to read and figures are hard to follow. Major improvements are needed before the paper can be published.

Thank you very much for the comments and suggestions, which improve our manuscript greatly. We have systematically revised the entire manuscript and conducted more clear and scientific conclusions. The corresponding modifications in the manuscript are marked in yellow. The followings are our specific responses to all the comments.

First two minor comments:

**Suggestion 1**. In the introduction, please give some examples of applications of the new model. It says that it can improve the positioning precision and enhance convergence speed, but of what and how much? Please mention some potential users of this model.

**Response 1**. Thank you for your suggestions. We have added detailed description of the positioning performance which is improved by the tropospheric model constraints in the introduction. The specific content is as follows:

==Xia et al. (2023) comprehensively considered the seasonal and intraday variations of the elevation normalization factor and developed a real-time ZTD model, and the vertical convergence speed was improved by 37.4% after the ZTD constraints are utilized to the float precise point positioning (PPP). Besides, FL-ZTD and SL-ZTD models are established using the piecewise exponential function as the key vertical adjustment scheme for ZTD, which reduced the convergence time by 60.0% and 33.3% compared to the standard PPP, respectively (Zhang et al., 2020). An optimized GPT3 model (RGPT3) is constructed using random forest (RF), achieving 12.3% and 7.9% improvement in vertical convergence speed and accuracy (Li et al., 2023).==

**Suggestion 2.** In several places acronyms appear without being defined. Please look through the manuscripts and change this.

**Response 2**. Thank you for your suggestions. We have carefully checked the manuscript and defined all abbreviations. The specific modifications are as follows:

==The zenith hydrostatic delay (ZHD)== can be accurately determined according to the Saastamoinen model with measured instantaneous pressure as the input, while the ==zenith wet delay (ZWD)== is generally estimated as an unknown parameter (Saastamoinen, 1972., Hadas et al., 2017., Zhang et al., 2021., Yang et al., 2023).

==The fifth generation European Centre for Medium-Range Weather Forecasts (ECMWF) atmospheric reanalysis (ERA5)== benefits from ==four-dimensional variational (4D-Var)== assimilation solution and ==integrated forecasting system (IFS)== forecast systems, which provides high spatial-temporal resolution and high-accuracy atmospheric state variables over globe (Hersbach et al., 2020; Jiang et al., 2023).

Atmospheric temperature, pressure and water vapor pressure data profiles at 0:00 ==coordinated universal time (UTC)== and 12:00 UTC in 2020 are obtained from the Integrated Global Radiosonde Archive (IGRA).

**Suggestion 3**. Then to the main problems with this paper, which are the figures and the presentation of the results. The panels are too small and provide too much information for the reader to digest. There is no need to display five almost completely blue panels like in figure 4. There is no need to display five almost completely blue panels like in figure 4.

**Response 3**. Thank you for your suggestions. We have simplified figure 4 and the corresponding analysis and only provided the mean accuracy of the four global schemes, which can also indicate that EFO3 is the optimal vertical fitting scheme.

The global mean and maximum root mean square (RMS) values of fitting residuals obtained by four solutions are shown in Figure 4. It is illustrated that the mean RMS of pressure fitted using EFO3 and EFO4 are less than 0.3 hPa on a global scale, they are clearly superior than those of EFO1 and EFO2. As for ZWD, the EFO2 outperforms EFO1, but the maximum RMS values still exceed 17 mm. The EFO3 generally performs identically to the EFO4, and their mean RMS values are less than 3.5 mm. As summarized above, the EFO1 and EFO2 can't reasonably account for the vertical characteristics of ZWD and pressure.

[Figure]

**Figure 4: The global mean and maximum RMS values for pressure (a) and ZWD (b) fitted using EFO1, EFO2, EFO3 and EFO4.**

**Suggestion 4**. Another example is that in figure 9 it is not possible to locate Greenland, Andes mountains or the Tibet Plateau which the authors refer to in the text. There are also both maps and bar charts in some of the figures where the bar charts are not explained. It would be beneficial to split these in to separate figures.

**Response 4**. Thank you for your suggestions. We have removed some of the subgraphs in section 3.2 due to excessive and complex information. The statistical results of bias and RMS are combined to more clearly and intuitively express the accuracy advantage of the proposed IGPZWD model in different height intervals and Temperature zones. Besides, the global statistical results are included in the Appendix A, which can effectively demonstrate the global and regional accuracy and applicability of IGPZWD model in different height ranges, providing indicators and references for users in different regions. The corresponding content in the revised manuscript is as follows:

The bias and RMS values of pressure predicted by GPT3, IGPT and IGPZWD models at three Temperature zones are presented in figure 12. It can be seen that the GPT3 model exhibits a systematic positive bias above 3 km, with a large mean bias value of 29 hPa in the temperate zone at the range of 12-15 km. Evidently, the accuracy of the GPT3 model gradually decreases with the increase of altitude, indicating that its pressure extrapolation scheme is inapplicable when the height difference is large. The IGPT model exhibits superior accuracy than the GPT3 model in the temperate and tropical regions where the intraday variations of pressure are strong, which benefits from the consideration of diurnal and semi-diurnal terms in pressure. IGPZWD model further effectively improves the accuracy

compared to IGPT model, achieving almost unbiased estimation of pressure with RMS improvements of 21.8-41.1% in tropical and 68.7-82.9% in temperate zones. In addition, the RMS of IGPZWD model has improved by over 94% compared to GPT3 model beyond 6 km in tropical regions, indicating the feasibility of the proposed vertical correction algorithm.

[Figure]

(a) Bias (hPa)  (b) RMS (hPa)

**Figure 12: Mean bias (a) and RMS (b) values (d1-d5) for pressure predicted by the GPT3, IGPT and IGPZWD models validated using the radiosonde pressure data at five height ranges of the tropical, temperate, and frigid zones.**

Table 1 summarizes the mean pressure bias and RMS values of each height range. The RMS values of IGPZWD do not exceed 5.3 hPa at five height ranges, showing a significant accuracy advantage compared to GPT3, with an improvement of up to 90% for 12-15 km. In contrast to the IGPT model, the IGPZWD model exhibits smaller negative bias values and further improves the performance beyond 3 km with RMS improvements of 32.4-51.8%, indicating the feasibility of the proposed vertical correction algorithm. The magnitude of ZWD in high altitude is small, and thus the pressure is the main factor restricting the accuracy of ZTD according to the rule of uncertainty propagation. It is implied that IGPZWD may provide superior prior tropospheric constraints for GNSS positioning of high-altitude platforms.

**Table 1: Mean bias and RMS values for pressure predicted by the GPT3, IGPT and IGPZWD models at five height ranges.**

| Height | Bias (hPa) | | | RMS (hPa) | | |
|---|---|---|---|---|---|---|
| (km) | GPT3 | IGPT | IGPZWD | GPT3 | IGPT | IGPZWD |
| 0-3 | 1.1 | -1.1 | 0.0 | 5.8 | 5.6 | 5.3 |
| 3-6 | 11.1 | -3.0 | -0.3 | 13.3 | 7.1 | 4.8 |
| 6-9 | 25.1 | -2.9 | -0.2 | 26.2 | 8.6 | 4.8 |
| 9-12 | 37.0 | -1.8 | -0.9 | 37.5 | 8.3 | 4.0 |
| 12-15 | 43.7 | -1.3 | -2.7 | 44.0 | 7.0 | 4.2 |

The bias and RMS values of pressure predicted by GPT3, IGPT and IGPZWD models at three Temperature zones are presented in figure 13. Significant negative bias values of the three models are observed in Southeast Asia below 3 km, which is attributed to the local strong annual and semi-annual amplitudes of ZWD. The GPT3 model exhibits generally positive bias values and large RMS values above 3 km in tropical and temperate zones, which again demonstrate that it can't provide reliable ZWD information in high-altitude areas. Although the GTrop model shows slight accuracy advantage below 3 km, while it performs worse than the IGPZWD model above 3 km. Compared to the GTrop model, IGPZWD model achieves RMS improvements of 14.5-27.8% and 10.6-48.5 % beyond 6 km in temperate and tropical zones, respectively, and the order of magnitude of improvement increases with height.

[Figure]

(a) Bias (mm)     (b) RMS (mm)

**Figure 13: Mean bias (a) and RMS (b) values (d1-d5) for pressure predicted by the GPT3, IGPT and IGPZWD models validated using the radiosonde ZWD data at five height ranges of the tropical, temperate, and frigid zones.**

**Suggestion 5**. My advice is to describe the over all (global) picture in the text and then show figures with some examples where there are big differences or interesting results. For example in figures 8 an 9, it would be more interesting to see full profiles of selected locations than 4 levels globally. This means a complete make over of sections 3 and 4.

**Response 5**. Thank you for your suggestions. We have added two sets of the experiment results in Section 3.1. We evaluated the vertical (1000 hPa-200 hPa) accuracies of each model in three representative regions and provided interesting conclusions and analysis. In addition, we retain relevant experiments and discussions of the four representative pressure levels over the globe. In this way, readers can simultaneously obtain the horizontal and vertical spatial applicability and accuracy information of the IGPZWD model through the four figures (8-11) and related analysis in Section 3.1. If necessary, further modifications and improvements can be made. The corresponding content in the revised manuscript is as follows:

Figure 9 depicts the vertical accuracies of pressure profiles predicted by GTP3, GTrop and IGPZWD models in three representative regions with different climatic environments and geographical locations. IGPZWD model exhibits overall optimal accuracy and stability with no significant sudden change. In the Tibet Plateau and Antarctic, the RMS and bias values of GPT3 model show evident and sharp trends of first decreasing and then increasing with altitude due to unreasonable pressure extrapolation method. Above 800 hPa, IGPT model tends to underestimate the pressure in the Andes mountains region, inducing systematic negative bias and relatively poorer RMS. Overall, the IGPZWD model achieves great pressure prediction on both the surface and the upper air, which benefits from the consideration of the seasonal variations for the pressure height scale factors.

[Figure]

(a) Tibet Plateau     (b) Andes mountains     (c) Antarctic

**Figure 9: Bias and RMS of pressure profiles predicted by the GPT3, IGPT, and IGPZWD models validated using the ERA5 pressure from 1000 to 200 hPa in 2020. The three selected regions are Tibet Plateau (a), Andes mountains (b) and Antarctica (c).**

Figure 11 illustrates that the GPT3 and GTrop models exhibit obviously positive bias in the Andes Mountains and Tibet Plateau below 800 hPa, and the RMS values of GPT3 exceeds 100 mm in the Tibetan Plateau region. In contrast, the IGPZWD model exhibits smaller bias values in these regions, and the RMS values are less than 40 mm. In the Antarctica, IGPZWD outperform all the other two models, achieving overall unbiased ZWD prediction above 400

hPa. It is concluded that IGPZWD model-predicted ZWD has a certain vertical accuracy advantage compared to GTrop and it is significantly more accurate than GPT3.

[Figure]

**Figure 11: Bias and RMS of ZWD profiles predicted by the GPT3, IGPT, and IGPZWD models validated using the ERA5 ZWD from 1000 to 200 hPa in 2020. The three selected regions are Tibet Plateau (a), Andes mountains (b) and Antarctica (c).**

**Suggestion 6**. No specific or technical comments will be given here since the paper needs such substantial work. This can be given at later stages of the review process.

**Response 6**. Thank you for your suggestions. We have systematically revised the article to demonstrate the feasibility and high accuracy of our model algorithm. Thank you again for your valuable suggestion.

---

## Author Comment (AC4)

**Response to Referee 2 Comments**

The authors describe a new model of pressure and zenith wet delay (ZWD) estimations focused on high-altitude areas. This work can be an interesting contribution to the field but the results are not properly represented and the validation is poorly explained. Please consider the following improvements:

Thank you very much for the comments and suggestions, which improve our manuscript greatly. We have systematically revised the entire manuscript and conducted more clear and scientific conclusions. The corresponding modifications in the manuscript are marked in yellow. The followings are our specific responses to all the comments.

**Suggestion 1**. Many acronyms are not defined

**Response 1**. Thank you for your suggestions. We have carefully checked the manuscript and defined all abbreviations. The specific modifications are as follows:

The zenith hydrostatic delay (ZHD) can be accurately determined according to the Saastamoinen model with measured instantaneous pressure as the input, while the zenith wet delay (ZWD) is generally estimated as an unknown parameter (Saastamoinen, 1972., Hadas et al., 2017., Zhang et al., 2021., Yang et al., 2023).

The fifth generation European Centre for Medium-Range Weather Forecasts (ECMWF) atmospheric reanalysis (ERA5) benefits from four-dimensional variational (4D-Var) assimilation solution and integrated forecasting system (IFS) forecast systems, which provides high spatial-temporal resolution and high-accuracy atmospheric state variables over globe (Hersbach et al., 2020; Jiang et al., 2023).

Atmospheric temperature, pressure and water vapor pressure data profiles at 0:00 coordinated universal time (UTC) and 12:00 UTC in 2020 are obtained from the Integrated Global Radiosonde Archive (IGRA).

**Suggestion 2**. Introduction should include studies about ZTD and its derivates. There are good examples in South America and Europe.

**Response 2**. Thank you for your suggestions. In the introduction, we have provided the description of precipitable water vapor (PWV) which can be derived from ZTD. The specific modifications are as follows:

Besides, the troposphere contains diverse atmospheric information. Accurate precipitable water vapor (PWV) can be derived by the combination of ZTD, atmospheric pressure and weighted mean temperature, and can be applied as an important indicator for regional and global numerical weather forecasting and meteorological monitoring (Wang et al., 2016; Li et al, 2022).

**Suggestion 3**. Figures do not represent the results. If the paper is focused on high-altitude zones, I recommend showing these areas in more detail.

**Response 3**. Thank you for your suggestions. We have added two sets of the experiment results in Section 3.1. We evaluated the vertical (1000 hPa-200 hPa) accuracies of each model in three representative regions (Tibet Plateau, Andes mountains and Antarctica) and provided interesting conclusions and analysis. In addition, we retain relevant experiments and discussions of the four representative pressure levels over the globe. In this way, readers can simultaneously obtain the horizontal and vertical spatial applicability and accuracy information of the IGPZWD model through the four figures (8-11) and related analysis in Section 3.1. The corresponding content in the revised manuscript is as follows:

Figure 9 depicts the vertical accuracies of pressure profiles predicted by GTP3, GTrop and IGPZWD models in three representative regions with different climatic environments and geographical locations. IGPZWD model

exhibits overall optimal accuracy and stability with no significant sudden change. In the Tibet Plateau and Antarctic, the RMS and bias values of GPT3 model show evident and sharp trends of first decreasing and then increasing with altitude due to unreasonable pressure extrapolation method. Above 800 hPa, IGPT model tends to underestimate the pressure in the Andes mountains region, inducing systematic negative bias and relatively poorer RMS. Overall, the IGPZWD model achieves great pressure prediction on both the surface and the upper air, which benefits from the consideration of the seasonal variations for the pressure height scale factors.

[Figure]

Figure 9: Bias and RMS of pressure profiles predicted by the GPT3, IGPT, and IGPZWD models validated using the ERA5 pressure from 1000 to 200 hPa in 2020. The three selected regions are Tibet Plateau (a), Andes mountains (b) and Antarctica (c).

Figure 11 illustrates that the GPT3 and GTrop models exhibit obviously positive bias in the Andes Mountains and Tibet Plateau below 800 hPa, and the RMS values of GPT3 exceeds 100 mm in the Tibetan Plateau region. In contrast, the IGPZWD model exhibits smaller bias values in these regions, and the RMS values are less than 40 mm. In the Antarctica, IGPZWD outperform all the other two models, achieving overall unbiased ZWD prediction above 400 hPa. It is concluded that IGPZWD model-predicted ZWD has a certain vertical accuracy advantage compared to GTrop and it is significantly more accurate than GPT3.

[Figure]

**Figure 11: Bias and RMS of ZWD profiles predicted by the GPT3, IGPT, and IGPZWD models validated using the ERA5 ZWD from 1000 to 200 hPa in 2020. The three selected regions are Tibet Plateau (a), Andes mountains (b) and Antarctica (c).**

**Suggestion 4**. There are no descriptions of the data sets used for validation beyond their names.

**Response 4**. Thank you for your suggestions. We have provided a more detailed introduction to the data. The corresponding content in the revised manuscript is as follows:

The fifth generation European Centre for Medium-Range Weather Forecasts (ECMWF) atmospheric reanalysis (ERA5) benefits from four-dimensional variational (4D-Var) assimilation solution and integrated forecasting system (IFS) forecast systems, which provides high spatial-temporal resolution and high-accuracy atmospheric state variables over globe (Hersbach et al., 2020). ERA5 provides 3D pressure-level products with a vertical resolution of 37 levels and 2D single-level data. The atmospheric parameters are provided with a horizontal resolution of 0.25°×0.25°, and the hourly data can more accurately reflect the short-term variation of meteorological parameters (Jiang et al., 2023). In this contribution, ERA5 hourly temperature, pressure, specific humidity and geopotential data from 2015 to 2019 are utilized to construct the IGPZWD model, and the accuracy of the new model is verified using data in 2020.

The Integrated Global Radiosonde Archive (IGRA) consists of radiosonde and pilot balloon observations from more than 2800 globally distributed stations, and surface and upper-air meteorological data become available in near real-time from about 800 stations worldwide (Ingleby et al., 2016). Atmospheric temperature, pressure and water vapor pressure data profiles at 0:00 coordinated universal time (UTC) and 12:00 UTC in 2020 are obtained from the IGRA.

**Suggestion 5**. Considering radiosonde data: Have you removed outliers? Did you apply a filter to exclude sites with a low amount of registers?

**Response 5**. Thank you for your suggestions. Strict quality control schemes have been applied to the radiosonde data to ensure its accuracy as reference values. We have added specific standards in the revised manuscript as follows:

Generally, sensor quality and weather events have a serious impact on raw measurements, which result in missing data and outliers. Hence, the low-quality radiosonde data profiles which meet the following quality control standards

are eliminated. (1) The height difference between two successive levels is greater than 2 km. (2) The pressure difference between two successive levels is greater than 200 hPa. (3) The height of the top-level data is less than 10 km. (4) The effective observation records of the profile are less than 20.

**Suggestion 6**. Furthermore, radiosondes in South America are few in comparison with North America and Europe, and many of them are located in low-altitude areas. Which criteria are used to describe the good fit of the model there?

**Response 6**. Thank you for your suggestion. We are sorry that we didn't clearly illustrate the verification method for radiosonde data, and our expression is not clear enough. The radiosonde data used in the study are the atmospheric pressure, ZWD and ZTD integration values of all observation records on the vertical profile at each radiosonde station below 15 km, not only the surface data. Therefore, all radiosonde stations can provide large vertical range profiles data and be regarded as reference values to evaluate the accuracy of models in high-altitude areas. Furthermore, we have simplified the images in section 3.2 to demonstrate the accuracy advantages of IGPZWD at different height ranges and Temperature zones, and added more detailed accuracy analysis and discussion for high-altitude areas. The corresponding content in the revised manuscript is as follows:

[revised manuscript text omitted]

**Suggestion 7**. If this paper wants to focus on height altitude sites, I recommend to analyze these areas separately.

**Response 7**. Thank you for your suggestions. Below 15 km, all observation records on the vertical profile of each radiosonde are used, which can provide large vertical range data and be regarded as reference values to evaluate the accuracy of models in high-altitude areas. We have systematically revised most of the content and added more results and discussions of high-altitude areas (3-15 km) to comprehensively demonstrate the wide spatial applicability and accuracy advantages of the proposed IGPZWD model. Thank you again for your valuable suggestion.

---

## Author Response (AR2)

**Response to Referee 1 Comments**

General comments:

In this paper, the authors developed the IGPZWD model, which can effectively provide global tropospheric parameters and outperform multiple advanced models. The models and methods show noteworthy innovation and potential application value, but there are some aspects that need further improvement.

Thank you very much for the suggestions, which improve our manuscript greatly. We have systematically revised the manuscript to meet the publication standards. The corresponding modifications in the manuscript are marked in yellow. The followings are our specific responses to all the comments.

The specific comments are as follows:

**Suggestion 1**. In section 2, please clarify what specific ERA5 parameters are used to establish the surface model and how the wet refractivity of single-level is calculated. Does the author use surface pressure, 2m-temperature, etc.?

**Response 1**. Thank you for your suggestions. We have added detailed description of the ERA5 single-level parameters for modeling and algorithm for single-level water vapor pressure which is further used to calculate surface wet refractivity in Section 2. The specific content is as follows:

In this contribution, ERA5 hourly temperature, pressure, specific humidity and geopotential data on pressure-level, surface pressure, 2m-dewpoint temperature and 2m-temperature on single-level from 2015 to 2019 are utilized to construct the IGPZWD model, and the accuracy of the new model is verified using data in 2020.

The ERA5 and radiosonde ZWD profiles are calculated according to the numerical integration method as follows (Thayer, 1974; Askne and Nordius, 1987; Jiang et al., 2023):

$$e = \begin{cases} PQ/(0.378 \times Q + 0.622) & Pressure - level \\ 6.112.\exp(17.62 \times Dew/(243.12 + Dew)) & Single - level \end{cases} \tag{1}$$

**Suggestion 2.** How to determine the vertical ranges of the fitting pressure and ZWD profiles in Figure 3 of Section 3.1, and why did the authors choose data profiles below 12 km instead of higher altitudes?

**Response 2**. Thank you for your suggestions. Pressure and ZWD exhibit significant vertical nonlinear (exponential) variations. Due to the different rate of decrease for atmospheric pressure in the upper troposphere compared to the lower troposphere, a fitting function with a single height scale factor cannot be used to characterize the vertical variation of atmospheric pressure throughout the neutral atmosphere. Besides, ZWD is almost zero beyond 10 km, so it's not meaningful to analyze the fitting effect of higher vertical ranges. Moreover, considering the vertical application space of airborne and high-altitude platforms, a height of 12 km is sufficient for use.

**Suggestion 3**. At the end of the section 3, it is recommended to insert a flowchart, so that readers can better understand the construction and application process of the IGPZWD model.

**Response 3**. Thank you for your suggestions. We have added a flowchart for the construction and use of IGPZWD model at the end of section 3. The corresponding content in the revised manuscript is as follows:

The development and use of the IGPZWD model are summarized in the flowchart depicted in Figure 8, including surface and vertical correction modules. With the geodetic location and time specified as DOY and HOD as inputs, the pressure and ZWD of the nearest four model grid points at the target height are determined according to equation

(17). Thereafter, a bilinear interpolation method is carried out to calculate the target pressure and ZWD. Furthermore, the target ZHD and ZTD are obtained based on the Saastamoinen model as follows:

[Figure]

**Figure 8: Flowchart depicting the development and use of the IGPZWD model.**

**Suggestion 4**. Tm gradually decreases with increasing altitude in the troposphere. The GPT3 model did not consider the vertical correction of Tm, thus overestimating the Tm value in high-altitude areas. However, this will result in a smaller ZWD value according to the empirical expression proposed by Askne and Nordius. Therefore, in section 4.1, the missing vertical correction for GPT3 model-derived Tm may not be the main reason for the systematic positive bias of ZWD. Please provide reasonable explanation.

**Response 4**. Thank you for your suggestions. According to Böhm et al, the water vapor decrease factor in the GPT3 model is calculated by water vapor pressure for any pair of pressure levels near the surface, thus only ensuring the accuracy of surface ZWD. In high-altitude areas, inaccurate decrease factors can cause significant extrapolation errors of water vapor pressure, leading to the accumulation of errors in ZWD. The corresponding content in the revised manuscript is as follows:

The statistical results of model-predicted ZWD validated using ERA5 profiles are shown in Figures 11 and 12. The magnitude of ZWD gradually decreases with increasing altitude, but the GPT3 model still shows a significant systematic positive bias at 350 hPa. This may be due to inaccurate estimation of the water vapor decrease factor, resulting in the accumulation of vertical errors. In contrast to GPT3 model, the GTrop and IGPZWD perform better at 550 and 350 hPa, showing smaller bias and RMS values in low latitudes.

**Suggestion 5**. Line 344, "Equation (20) is essentially based on the assumption of isothermal atmosphere". - replace (20) by (19).

**Response 5**. Thank you for your suggestions. We have made corresponding modifications and carefully checked all sections to avoid such errors. The corresponding content in the revised manuscript is as follows:

Equation (19) is essentially based on the assumption of isothermal atmosphere, but the actual atmospheric state does not meet the condition, except for the tropopause.

**Suggestion 6**. Please clarify the shortcoming of the new model in the conclusion.

**Response 6**. Thank you for your suggestions. We revealed the shortcomings of the IGPZWD model. The corresponding content in the revised manuscript is as follows:

Despite the complexity of the inversion process for tropospheric parameters, the overall performance of IGPZWD model is encouraging. In the future, the model coefficients will be further simplified to balance the computation efficiency and accuracy.

**Response to Referee 2 Comments**

This manuscript presents an improved global pressure and ZWD model that takes into account the spatiotemporal variability of multiple height scale factors, as well as its optimized vertical correction work, which has positive value for GNSS motion accuracy positioning and atmospheric water vapor detection. This manuscript is a revised version and has been carefully revised based on the comments of the reviewers. Regarding the revised manuscript, I have only two suggestions.

Thank you for your gradual processing and valuable suggestions, which has once again improved the quality of this paper. We have further revised the discussion, conclusions and outlook to better demonstrate the value and significance of this contribution. The corresponding modifications in the manuscript are marked in yellow. The followings are our specific responses to all the comments.

**Suggestion 1**. The outlook section should be included in the discussion section, not in the conclusion section.

**Response 1**. Thank you for your suggestions. We have simplified the conclusion section and included the outlook section in the discussion. The specific modifications are as follows:

**Line 395:**

To sum up, the proposed IGPZWD model can provide high-quality tropospheric parameters prediction below 15 km on a global scale. The IGPZWD model will be of great significance for the tropospheric augmentation in real-time GNSS positioning, and it has broad application prospects in real-time water vapor sounding and extreme weather forecasting. Despite the complexity of the inversion process for tropospheric parameters, the overall performance of IGPZWD model is encouraging. In the future, the model coefficients will be further simplified to balance the computation efficiency and accuracy.

**Line 420:**

In summary, the proposed optimized vertical correction algorithm weakens the cumulative error caused by large correction height difference, which effectively improves the accuracy and stability of pressure, ZWD and ZTD in high-altitude areas.

**Suggestion 2**. The discussion section is weak, and it is suggested that the author should combine the excellent research references of predecessors in this field more, rather than just talking to themselves. In summary, it is recommended to undergo moderate revision.

**Response 2**. Thank you for your suggestions. We carefully revised the discussion section and combined the analysis and conclusions of other studies to further evaluate the algorithm advantages and feasibility of the IGPZWD model. We hope that our comprehensive assessment can provide important reference value for the application of scholars. The specific modifications are as follows:

**Line 290:**

Figure 12 illustrates that the GPT3 and GTrop models exhibit obviously positive bias in the Andes Mountains and Tibet Plateau below 800 hPa, and the RMS values of GPT3 exceeds 100 mm in the Tibetan Plateau region. In contrast, the IGPZWD model exhibits smaller bias values in these regions, and the RMS values are less than 40 mm. In the Antarctica, IGPZWD outperform all the other two models, achieving overall unbiased ZWD prediction above 400 hPa. It is concluded that IGPZWD model-predicted ZWD has a certain vertical accuracy advantage compared to GTrop and it is significantly more accurate than GPT3. Although IGPZWD-predicted ZWD exhibit superior performance in high-altitude areas, the improvement in surface is negligible. It is concluded that developing surface

ZWD models is challenging. Nevertheless, substantial studies have proven that the cubic polynomial can effectively improve the fitting effect of ZWD profiles at low altitudes, which can be the algorithm reference for future enhanced model construction (Li et al., 2023; Xu et al., 2023).

**Line 315:**

In addition, the RMS of IGPZWD model has improved by over 94% compared to GPT3 model beyond 6 km in tropical regions, indicating the feasibility of the proposed vertical correction algorithm. In summary, it has been verified that exponential function with three orders has stronger accuracy advantages and robustness in vertical pressure extrapolation compared to existing models such as virtual temperature and the adiabatic models. In addition, we further achieved accurate prediction of the height scale factor time series for pressure.

**Line 340:**

Compared to the GTrop model, IGPZWD model achieves RMS improvements of 14.5-27.8% and 10.6-48.5 % beyond 6 km in temperate and tropical zones, respectively. And the order of magnitude of improvement increases with height, confirming the advantages of high-order exponents in simulating ZWD profiles. Previous scholars have used piecewise and high-order functions to characterize the vertical nonlinear variation of ZWD (Hu and Yao, 2019; Zhu et al., 2022). We have also verified the feasibility of this kind of algorithm using 1-year radiosonde data. Moreover, it has been proven that considering time-varying height scale factor can further improve the long-term forecasting accuracy.